# The Long Noncoding RNA LOC441461 (STX17-AS1) Modulates Colorectal Cancer Cell Growth and Motility

**DOI:** 10.3390/cancers12113171

**Published:** 2020-10-28

**Authors:** Jui-Ho Wang, Tzung-Ju Lu, Mei-Lang Kung, Yi-Fang Yang, Chung-Yu Yeh, Ya-Ting Tu, Wei-Shone Chen, Kuo-Wang Tsai

**Affiliations:** 1Division of Colorectal Surgery, Department of Surgery, Kaohsiung Veterans General Hospital, Kaohsiung 81362, Taiwan; wang@vghks.gov.tw; 2Division of Colon and Rectal Surgery, Department of Surgery, Taipei Tzu Chi Hospital, Buddhist Tzu Chi Medical Foundation, New Taipei City 23124, Taiwan; ndmcnalien@gmail.com; 3Department of Medical Education and Research, Kaohsiung Veterans General Hospital, Kaohsiung 81362, Taiwan; kungmeilang@gmail.com (M.-L.K.); yvonne845040@gmail.com (Y.-F.Y.); cyyeah@vghks.gov.tw (C.-Y.Y.); 4Department of Research, Taipei Tzu Chi Hospital, Buddhist Tzu Chi Medical Foundation, New Taipei City 23124, Taiwan; new0224@hotmail.com; 5School of Medicine, National Yang-Ming University, Taipei 11221, Taiwan; wschen@vghtpe.gov.tw; 6Department of Surgery, Veterans General Hospital, Taipei 11221, Taiwan

**Keywords:** lncRNA, LOC441461, colon cancer

## Abstract

**Simple Summary:**

Long noncoding RNA dysfunction is crucial for colorectal carcinoma (CRC) development. Whether the dysfunction of LOC441461, a novel lncRNA, can regulate cancer-related signaling pathways in cancer progression remains unclear. Here, we uncover the oncogenic role of LOC441461 in colon cancer cell growth and motility and identify a novel mechanism for LOC441461 knockdown-induced suppression of cancer motility through modulating Ras homolog family member A (RhoA)/Rho-associated protein kinase (ROCK) activity. This is the first report that LOC441461 knockdown impairs cell cycle progression and accelerates the apoptosis of colon cancer cells following chemotherapy drug treatment. The results suggest that LOC441461 expression confers drug sensitivity in colon cancer by inducing apoptosis. Our findings offer new insight into LOC441461 regulation and provide an application for colon cancer therapy in the future.

**Abstract:**

Colorectal carcinoma (CRC) is one of the most prevalent cancers worldwide and has a high mortality rate. Long noncoding RNAs (lncRNAs) have been noted to play critical roles in cell growth; cell apoptosis; and metastasis in CRC. This study determined that LOC441461 expression was significantly higher in CRC tissues than in adjacent normal mucosa. Pathway enrichment analysis of LOC441461-coexpressed genes revealed that LOC441461 was involved in biological functions related to cancer cell growth and motility. Knockdown of the LOC441461 expression significantly suppressed colon cancer cell growth by impairing cell cycle progression and inducing cell apoptosis. Furthermore, significantly higher LOC441461 expression was discovered in primary colon tumors and metastatic liver tumors than in the corresponding normal mucosa, and LOC441461 knockdown was noted to suppress colon cancer cell motility. Knockdown of LOC441461 expression suppressed the phosphorylation of MLC and LIMK1 through the inhibition of RhoA/ROCK signaling. Overall, LOC441461 was discovered to play an oncogenic role in CRC cell growth and motility through RhoA/ROCK signaling. Our findings provide new insights into the regulation of lncRNAs and their application in the treatment of colon cancer

## 1. Introduction

Colorectal carcinoma (CRC) is one of the most prevalent cancers worldwide and has a high mortality rate [1,2]. Although the mortality rate of CRC has gradually decreased, its overall survival rate remains poor when the cancer is advanced. Notably, in the advanced stage of CRC, metastasis and drug resistance are major problems that can be fatal [3]. Determining the molecular mechanism of colon cancer genesis and metastasis is a research topic of interest. Therefore, the identification and elucidation of the dysregulated genes involved in colon cancer progression and the development of an excellent biomarker to aid early diagnosis or act as a potential therapeutic target would benefit patients with CRC.

Noncoding RNAs (ncRNAs) are functional RNA transcripts that lack protein translation ability, and ncRNA dysfunction plays a crucial role in human cancer progression [4,5]. On the basis of their size, ncRNAs can be classified as small ncRNAs (microRNAs) or long noncoding RNAs (lncRNAs). LncRNAs are RNA transcripts that have a length of >200 nucleotides. Increasing evidence has revealed that lncRNA dysfunction plays a critical role in CRC carcinogenesis, cancer cell growth, and metastasis [6,7,8]. Studies have reported that numerous lncRNAs such as HOTAIR, PCAT-1, PVT-1, H19, BANCR, 91H, CCAT1, MALAT1, LINC00504, B4GALT1-AS1, and Linc00659 are dysregulated in colon cancer [7,8,9,10,11,12,13,14,15,16,17,18,19,20,21]. Our previous study identified several dysregulated lncRNAs in CRC by using the microarray approach [7]. The biological function of LOC441461 in human cancer cells remains unclear. LOC441461 shares a bidirectional promoter with STX17 at human chromosome 9. Studies have indicated that antisense ncRNA positively or negatively regulates the expression of sense protein–coding genes [22,23,24,25]. In this study, we assessed the expression levels of STX17 and LOC441461 in colon cancer. We determined that only LOC441461 was significantly overexpressed in colon cancer compared with adjacent normal tissues. Furthermore, our findings revealed that LOC441461 has a novel oncogenic role in regulating CRC cell growth and migration through modulating RhoA/ROCK signaling and can be a target for gene therapy.

## 2. Results

### 2.1. Expression Levels of LOC441461 Were Significantly Increased in CRC

In our previous study, we determined the lncRNA profile of CRC by using the microarray approach and identified several lncRNAs that were deregulated in CRC tissues compared with adjacent normal tissues [7]. The role of one of these lncRNAs in CRC, namely LOC441461, remains unclear. Notably, LOC441461 is a 553-bp-long ncRNA that shares a bidirectional promoter with STX17 at human chromosome 9:99,886,317–99,906,601 (Figure 1A). We further determined the RNA transcript of LOC441461 by using polymerase chain reaction (PCR) and the Sanger sequencing approach. Our data indicated that the RNA transcript of LOC441461 consists of three exons (Appendix A). Microarray data revealed that LOC441461 was upregulated (more than threefold) in CRC compared with adjacent normal tissue, whereas the expression level of STX17 was unchanged in CRC (Figure 1B,C). We further examined the expression levels of LOC441461 and STX17 in colon cancer by analyzing The Cancer Genome Atlas (TCGA) database, which revealed that LOC441461 was significantly upregulated in colon cancer compared with adjacent normal tissues (*p* = 0.0019). By contrast, no difference was discovered in STX17 expression between colon cancer and normal tissues (*p* = 0.95; Figure 1D,E). We further examined the expression levels of LOC441461 by using real-time (RT)-PCR, which revealed that LOC441461 expression was significantly increased in colorectal cancer compared with adjacent normal mucosa (in tissues from 70 out of 89 patients; Figure 1F).

### 2.2. LOC441461 Expressed with Cancer-Related Signaling Pathway Dysfunction

We also identified a group of genes with positive and negative coexpression with LOC441461 in CRC to explore the putative function. We downloaded the RNA transcriptome of 41 N-T pairs of patients with CRC from TCGA database. By calculating the correlation between the expression of LOC441461 and protein-coding genes in CRC, the negatively and positively coexpressed gene candidates were identified. Overall, 200 gene candidates, 100 with positive correlations and 100 with negative correlations with LOC441461 expression, were selected for further analysis, which revealed that 35 coexpressed genes were significantly upregulated and 77 coexpressed genes were significantly downregulated in CRC (Figure 2A,B). These differentially expressed genes were subjected to pathway enrichment analysis by using DAVID Bioinformatics Resources 6.8 (https://david.ncifcrf.gov/). As illustrated in Figure 2B, the positively coexpressed genes were significantly enriched in targeting mitochondria, microtubule anchoring, and the Notch signaling pathway, whereas the downregulated genes were significantly involved in cell shape regulation, small GTPase regulation, mitotic nuclear division, and protein localization to the preautophagosomal structure. Gene ontology analysis of all differentially expressed genes revealed that these genes were significantly involved in protein targeting of mitochondria, protein transport, cell shape regulation, intracellular protein transport, cellular response to nerve growth factor stimulus, regulation of GTPase activity in biological processes, transferrin transport, coat protein complex I (COPI) coating of Golgi vesicles, positive regulation of cholesterol storage, cellular response to laminar fluid shear stress, macropinocytosis, regulation of Golgi organization, and G2/M transition of the mitotic cell cycle (Appendix A).

### 2.3. LOC441461 Regulated Colon Cancer Cell Growth by Impairing Cell Cycle Progression

Pathway enrichment analysis revealed that genes coexpressed with LOC441461 were significantly involved in the cancer-related signaling pathway, especially in the pathway regulating the cell cycle, cell shape, and mitochondrion structure. This result implies that LOC441461 may participate in colon cell growth, apoptosis, and migration. Nevertheless, to explore the role of LOC441461 in colon cancer, we measured its expression levels by using RT-PCR in eight colon cancer cell lines: colo320DM, colo205, DLD-1, LoVo, SW620, HCT116, LS174T, and HT29. The LOC441461 expression level was high in colo320DM, colo205, and DLD-1; moderate in LoVo, SW620, HCT116, and LS174T; and low in HT29 cells, as illustrated in Figure 3A. Furthermore, we analyzed the subcellular localization of LOC441461, which revealed that LOC441461 expression occurred predominantly in the cytoplasm of colon cancer cells (Appendix A). First, we examined the effect of LOC441461 on colon cancer cell growth by knocking down its expression in SW620 cells by using the small interfering RNA (siRNA) approach. Overall, three siRNA oligonucleotides were designed to target the LOC441461 gene, namely si-LOC441461_318, si-LOC441461_384, and si-LOC441461_432, which could silence LOC441461 expression by at least 50% after transfection with individual siRNA for 48 h (Appendix A). The knockdown of LOC441461 with individual siRNA suppressed SW620 cell growth (Appendix A). Moreover, we pooled three siRNAs and reduced the concentration employed to prevent the off-target effect. The LOC441461 expression level decreased by 70% in the SW620 cells subjected to si-LOC441461-pool (si-LOC441461_318 + si-LOC441461_384 + si-LOC441461_432) transfection for 48 h, as illustrated in Figure 3B. Furthermore, LOC441461 knockdown significantly suppressed cell colony formation and anchorage-independent growth in SW620 cells (Figure 3C,D). Knockdown of the LOC441461 expression suppressed SW620 cell proliferation (Figure 4A). The cell cycle distribution of SW620 cells with LOC441461 knockdown was examined to identify the detailed mechanism through which LOC441461 knockdown inhibited SW620 cell growth. An image flow assay revealed that the G0/G1 phase was significantly increased in the SW620 cells subjected to si-LOC441461 transfection (Figure 4B,C). Furthermore, cell-cycle-related protein expression revealed that the SW620 cells with LOC441461 knockdown had decreased cyclin A, B, and D levels, but increased CDKNIA expression (Figure 4D and Appendix A). Similar results of LOC441461-knockdown-induced colon cancer growth suppression were observed in the LoVo, LS174T, and DLD-1 cell lines (Appendix A). Moreover, the number of cells in the G0 and G1 phases was marginally higher in the LoVo, LS174T, and DLD-1 cell lines with LOC441461 knockdown than in the control cell line (Appendix A).

Our data also revealed that LOC441461 knockdown induced colon cancer cell apoptosis (Figure 4E,F). We assessed whether LOC441461 expression contributed to drug responsiveness and examined the effects of LOC441461 knockdown on cell apoptosis in SW620 cells subjected to drug treatment. As depicted in Figure 4E,F, LOC441461 knockdown accelerated the apoptosis of SW620 cells following treatment with oxaliplatin, 5-FU, or irinotecan. This result suggested that LOC441461 expression contributes to the drug sensitivity of colon cancer cells.

### 2.4. LOC441461 Involved in Colon Cancer Motility

Notably, DLD-1 cells were more invasive than LoVo, LS174T, and HT29 cells, as illustrated in Figure 5A–D. The LOC441461 expression level was high in DLD-1 cells, moderate in LoVo and LS174T cells, and low in HT29 cells (Figure 3A). Furthermore, pathway enrichment analysis revealed that LOC441461-coexpressed genes were involved in microtubule anchoring and cell shape regulation (Figure 2B). This result implied that LOC441641 may be involved in the regulation of colon cancer cell migration and invasion. The knockdown of LOC441461 expression significantly suppressed the invasion, migration, and wound healing abilities of DLD-1 cells (Figure 5E,F and Appendix A). The Twist expression level in DLD-1 cells with LOC441461 knockdown was lower than that in the control cells (Figure 5G and Appendix A). We also examined the effects of LOC441461 knockdown on motility in LoVo and LS174T cells and discovered that LOC441461 knockdown suppressed the invasion and migration abilities of LoVo cells but not that of LS174T cells (Appendix A). Furthermore, we examined the LOC441461 expression level in the adjacent normal tissues, primary CRC, and liver metastases of 35 patients with CRC. Our data indicated that LOC441641 expression was significantly higher (*p* < 0.001, Figure 5H) in CRC than in adjacent normal mucosa. Furthermore, higher levels of LOC441461 expression were discovered in the metastatic liver tumors than in the primary colon tumors (*p* = 0.04, Figure 5H)

Our results revealed that LOC441461 knockdown suppressed colon cancer cell growth and motility. According to pathway enrichment analysis, LOC441461-coexpressed genes were significantly involved in regulating the small GTPase activity, cell shape, and cell cycle. Studies have reported that the Rho family of small GTPases regulates crucial cellular processes, including cytoskeletal dynamics and cell migration and growth [26,27]. Therefore, we suggest that LOC441461 knockdown suppresses cancer cell growth and motility by blocking RhoA/ROCK signaling in colon cancer. We further examined the RhoA/ROCK/MLC2 and RhoA/ROCK/LIMK signaling in DLD-1 cells with LOC441461 knockdown. As illustrated in Figure 5I, RhoA, ROCK, and MLC expression levels were reduced in the DLD-1 cells with LOC441461 knockdown. Our data also revealed that LOC441461 knockdown could suppress the phosphorylation of MLC and LIMK1 (Figure 5I and Appendix A). Furthermore, LOC441461 knockdown reduced the generation of cell membrane filopodia protrusions in the DLD-1 cells (Appendix A). In the aforementioned cells, the amount of G-actin (monomer) increased, whereas that of F-actin decreased (Appendix A). In summary, our study is the first to report the involvement of a novel oncogenic lncRNA, namely LOC441461, in colon cancer growth and cell motility through the modulation of the RhoA/ROCK signaling activity (Figure 5J).

## 3. Discussion

This study identified a novel oncogenic lncRNA, LOC441461, involved in colon cancer cell growth and cell invasion. However, the detailed mechanisms of LOC441461-related cell growth and motility are unclear. The subcellular localization of LOC441461 was also analyzed and revealed higher expression levels of LOC441461 in the cytoplasm than in the nucleus (Appendix A). Thus, when its location is considered, LOC441461 may regulate colon cancer cell growth by binding with microRNA or modulating translation to prevent tumor suppressor gene expression. Tavazoie et al. [28] reported that miR-335 inhibited metastatic breast cancer invasion by regulating a set of genes, including SOX4 and extracellular component tenascin. On searching the Gene Expression Omnibus (GDS3138) (https://www.ncbi.nlm.nih.gov/geoprofiles/?term=GDS3138), we noted that LOC441461 expression was downregulated in LM2 cells with miR-335 overexpression. This result implied that LOC441461 may regulate the growth and motility of colon cancer cells by sponging miR-335. However, further research is warranted to determine the detailed mechanism through which LOC441461 has an effect on the growth and motility of colon cancer cells.

LOC441461 is an antisense RNA and shares a bidirectional promoter with the STX17 gene in the human genome. Furthermore, the LOC441461 expression level was significantly higher in colon cancer cells than in adjacent normal tissues, whereas no differences were observed in the STX17 expression between colon cancer cells and adjacent normal tissues. Therefore, we concluded that LOC441461 and STX17 share a bidirectional promoter but are not coexpressed in colon cancer. STX17 is a localized endoplasmic reticulum membrane protein that interacts with ATG14L, Fis1, and BAP31 in the formation of a functional complex to modulate cell survival [29,30,31]. Studies have indicated that STX17 is a TBK1 substrate that participates in the assembly of protein complexes during autophagy initiation [32,33]. Huang et al. [34] reported that MALAT1 regulated the autophagy of retinoblastoma cells through miR-124-mediated STX17 regulation. Nonetheless, no information is yet available regarding the involvement of LOC441461 in human cancer. Moreover, its biological role is unknown. First, we determined that LOC441461 plays an oncogenic role by promoting colon cancer proliferation and motility. Then, we identified LOC441461-coexpressed genes by using RNA transcriptome data from TCGA. We attempted to determine the mechanism of action and determined that these coexpressed genes were significantly involved in protein targeting of mitochondria, protein transport, Golgi organization, and determining the cell shape and cell cycle. In addition, mitochondria-dependent apoptosis was revealed to play a critical role in human cancer progression, including colon cancer [35]. Moreover, the Golgi structure is essential for the efficient processing of mammalian biological functions, including apoptosis [36]. Similar to mitochondria, the Golgi complex can act as a sensor of proapoptotic signals through local caspase 2 [37]. The LOC441461-coexpressed genes were observed to be significantly involved in mitochondria and Golgi structure modulation, and LOC441461 knockdown suppressed CRC cell growth by inducing cell apoptosis. The aforementioned results revealed that LOC441461 may participate in the modulation of the mitochondria or Golgi apparatus to regulate cancer cell growth.

Pathway enrichment analysis revealed that LOC441461-coexpressed genes were significantly involved in regulating small GTPase activity, cell shape, and cell cycle. In human malignancies, most Rho GTPases are aberrantly expressed and contribute to the regulation of cancer cell proliferation, metastasis, and angiogenesis. Knockdown of RhoA expression significantly suppressed cancer cell growth and tumorigenesis and enhanced the chemosensitivity of cancer cells to treatment with Adriamycin and 5-fluorouracil [38]. Zhang et al. [39] reported that the blocking of the Rho-ROCK pathway impaired the cell cycle G1–S transition due to the increase in the P21(waf1/Cip1) and p27(Kip1) expression and decrease in the activities of CDK4 and CDK6. Furthermore, cell-cycle-dependent Rho GTPase activity was shown to regulate cancer cell migration and invasion dynamically. A Rho-GTPase-activating protein, namely ARHGAP11A, was expressed in a cell-cycle-dependent manner and induced cell cycle arrest through interaction with p53 [40]. Interestingly, ARHGAP11A expression induced an increase in the relative Rac1 activity by blocking RhoA signaling, which led to an increase in the invasion ability of colon cancer cells [41]. Studies have demonstrated that microRNA and lncRNA modulate the growth and invasion properties of human cancer cells through the fine-tuning of RhoA/ROCK signaling [42,43,44]. Scholars have reported numerous lncRNAs involved in regulating human cancer cell growth and metastasis through the modulation of the RhoA pathway, including SNHG5, LOC554202, and MALAT1 [26,45]. In the present study, we discovered a novel oncogenic lncRNA (i.e., LOC441461) that regulated the growth and motility of colon cancer cells and conferred sensitivity through the RhoA-ROCK signaling activity. Our data indicated that LOC441461 knockdown suppressed colon cancer growth by inducing cell arrest in the G1 phase and providing drug sensitivity to SW620 cells. Our data indicated that LOC441461 knockdown induced cell cycle arrest in the G0/G1 phase. However, the numbers of cells in the G0 and G1 phases was only marginally higher in the LoVo, LS174T, and DLD-1 cell lines with LOC441461 knockdown than in the control group cell line (Appendix A). Although LOC441461-knockdown-induced suppression of CCNB1 was observed in all colon cancer cells, the suppression of CCND1 and CDK4 was observed in SW620, DLD-1, and LS174T cells but not LoVo cells (Appendix A). The marginally inconsistent aforementioned results may be related to the different genetic backgrounds of distinct colon cancer cells.

## 4. Materials and Methods

### 4.1. Colon Cancer Cell Line

In this study, eight human CRC cell lines were obtained from the American Type Culture Collection (https://www.atcc.org/en/Products/Cells_and_Microorganisms/Cell_Lines.aspx): DLD-1, Colo205, colo320DM, LS174T, HCT116, SW620, LoVo, and HT29. These cells were cultured in high-glucose Dulbecco’s Modified Eagle’s Medium (Invitrogen, Grand Island, NY, USA) supplemented with 10% fetal bovine serum (FBS; Hyclone Laboratories Inc., South Logan, UT, USA) and 1% penicillin–streptomycin (Gibco, Thermo Fisher Scientific Inc., Waltham, MA, USA) in plastic tissue-culture plates at 37 °C under a humidified atmosphere containing 5% CO_2_.

### 4.2. Clinical Samples

Primary colon cancer tissue and adjacent normal mucosa samples were obtained from the biobanks of Kaohsiung Veterans General Hospital and Taipei Tzu Chi Hospital, Taiwan. In addition, liver tumor, primary tumor, and adjacent normal mucosa samples were obtained from 35 patients with CRC who underwent surgery at the Department of Surgery, Taipei Veterans General Hospital, Taiwan. All the data and specimens used in this study were obtained from the biobank of a hospital and anonymously analyzed. Informed consent was obtained from all patients by the biobanks. This study was approved by the ethics committees of Kaohsiung Veterans General Hospital (VGHKS15-CT8-26), Taipei Tzu Chi Hospital (09-X-008), and Taipei Veterans General Hospital (IRB number: 2015-06-012CC).

### 4.3. Ethics Approval and Consent to Participate

This study was approved by the ethics committees of Taipei Veterans General Hospital, Kaohsiung Veterans General Hospital, and Taipei Tzu Chi Hospital. Moreover, written informed consent was obtained from all patients (IRB number: 09-X-008, 2015-06-012CC, VGHKS15-CT8-26).

### 4.4. Extraction of RNA and Reverse Transcription

The total RNA of clinical tissues or cell lines was extracted using TRIzol reagent (Invitrogen, Grand Island, NY, USA). The clinical samples were homogenized using 1 mL of TRIzol reagent; 0.2 mL of chloroform was then added to extract protein. RNA was precipitated using 0.5 mL of isopropanol, and the pellet was then eluted using 20 μL of H_2_O. The concentration and purity of the total RNA were determined using the Nanodrop 1000 spectrophotometer (Nanodrop Technologies Inc., Wilmington, DE, USA). For the reverse transcription reaction, 2 µg of total RNA, random primers (Invitrogen, Grand Island, NY, USA), and SuperScript IV Reverse Transcriptase (Invitrogen, Grand Island, NY, USA) were used for cDNA conversion according to the manufacturer’s instructions (Invitrogen, Grand Island, NY, USA). The reaction was performed through incubation at 42 °C for 1 h, and the enzyme was subsequently inactivated by incubation at 85 °C for 5 min.

### 4.5. RT-PCR

The obtained cDNA was used for RT-PCR analysis with LOC441461- and glyceraldehyde 3-phosphate dehydrogenase (GAPDH)-specific primers, and gene expression was detected using the Fast SYBR Green Master Mix (Applied Biosystems; Thermo Fisher Scientific Inc. Waltham, MA, USA). RT-PCR was performed under the following conditions: incubation at 94 °C for 10 min, followed by 40 cycles of incubation at 94 °C for 15 s and at 60 °C for 32 s. Finally, the expression of LOC441461 was normalized to that of GAPDH (Δ*Ct* = gene *Ct* − GAPDH *Ct*). The individual primers used in this study were as follows:

LOC441461-F: 5′-TGATAAGCTGTTTAAACCAGAACCG-3′;

LOC441461-R: 5′-GGCAACATTTCAGTTCCAGTG-3′;

GAPDH-F: 5′-TGCACCACCAACTGCTTAGC-3′; and

GAPDH-R: 5′-GGCATGGACTGTGGTCATGAG-3′.

### 4.6. Expression Data from TCGA

Transcriptome expression data of colon cancer were downloaded from TCGA data portal (https://tcga-data.nci.nih.gov/tcga/dataAccessMatrix.htm). The expression profiles of 444 colon cancer tissues and 41 adjacent normal tissues were obtained from TCGA data portal. In this study, the correlation between LOC441461 and protein-coding genes in colon cancer tissues from 41 patients was assessed using Pearson correlation. The 100 gene candidates with the strongest negative and positive correlations with LOC441461 were further examined in N-T paired colon cancer tissues from 41 patients, and the differentially expressed gene candidates in CRC were identified at the significance level *p* < 0.05.

### 4.7. Pathway Enrichment Analysis

Differentially expressed genes were subjected to gene ontology analysis by using DAVID Bioinformatics Resources 6.8 [46] to identify significantly enriched pathways.

### 4.8. siRNA Transfection

LOC441461 knockdown was performed using an siRNA oligonucleotide pool (si-LOC441461-318, sense: 3′–5′-GGAACUGAAAUGUUGCCUUTT-3’, antisense: 5′-AAGGCAACAUUUCAGUUCCTT-3′; si-LOC441461-432, sense: 3′–5′-GCUGCUACAUUAACUGAUUTT-3′, antisense: 5′-AAUCAGUUAAUGUAGCAGCTT-3′; and si-LOC441461-384, sense: 3′–5′-GAUGGACUCACUAAAGGAUTT-3′, antisense: 5′-AUCCUUUAGUGAGUCCAUCTT-3′) and a scrambled oligo as a negative control. The pool and oligo were designed and synthesized by GenDiscovery Biotechnology (Taipei, Taiwan). Cells were transfected with a final concentration (10 mM) of individual siRNA or control by using Lipofectamine RNAiMAX reagent (Invitrogen; Thermo Fisher Scientific Inc. Waltham, MA, USA). After transfection for 24 h, RNA was extracted, and the knockdown efficiency was evaluated using specific primers by performing RT-PCR.

### 4.9. Proliferation

For cell proliferation analysis, 2500 living cells were transfected with si-LOC441461 or scrambled control and were plated onto 96-well plates. Cell growth was determined on days 0, 1, 2, 3, and 4. Cell viability was determined using a CellTiter-Glo One Solution cell proliferation assay (Promega Corporation, Madison, WI, USA).

### 4.10. Soft Agar Assay

The base agar layer was prepared with 1% agar (Laboratorios CONDA, Torrejón de Ardoz, Madrid, Spain) dissolved in 1.5 mL of phosphate-buffered saline (PBS) in six-well plates. Subsequently, 1.5 mL of 0.7% agarose (Invitrogen, Grand Island, NY, USA) solution containing the cells (15,000 cells per well) was inoculated on top of the base agar layer. After allowing the solution to harden, 0.5 mL of fresh medium was added to the top of the hard agar layer. After 3–4 weeks, the agar plates were stained with iodonitrotetrazolium chloride (Santa Cruz Biotechnology, Santa Cruz, CA, USA) and the colonies were counted.

### 4.11. Colony Formation Assay

Overall, 4000 cells were seeded in each well of a six-well plate, and the cells were transfected with si-LOC441461 or scrambled control by using the Lipofectamine RNAiMAX reagent (Invitrogen, Grand Island, NY, USA) in the six-well plate. After incubation at 37 °C for 3 days, the cultured medium was replaced with fresh medium. The cells were then incubated at 37 °C for 10 days. Cell culture plates containing colonies were fixed with 4% formaldehyde for 2 min, and colonies were stained using crystal violet solution (containing 0.5% crystal violet, 5% formaldehyde, 50% ethanol, and 0.85% sodium chloride) for 2 h. Wells were rinsed with water after air drying. The crystal violet staining of the cells from each well was solubilized using 1 mL of 10% acetic acid per well, and the absorbance (optical density) of the solution was measured using a spectrophotometer at a wavelength of 595 nm.

### 4.12. Cell Cycle Analysis

Overall, 1 × 10^6^ cells were collected and mixed with 70% ethanol in a fixative at −20 °C overnight. The cells were then stained with 4′,6-diamidino-2-phenylindole (ChemoMetec, Gydevang, Lillerød, Denmark) and detected using the NucleoView NC-3000 software (ChemoMetec).

### 4.13. Apoptotic Cells Stained with Annexin V, Propidium Iodide, and Hoechst 33342

Apoptotic cells were detected using the CF488A Annexin V and PI Apoptosis kit (Biotium, Fremont, CA, USA) and Hoechst 33342 (ChemoMetec). After staining, the cells were analyzed using an image flow cytometry assay (NucleoCounter NC-3000, ChemoMetec). NucleoView NC-3000 was then employed for automated image flow cytometry analysis.

### 4.14. Invasion and Migration Assay

The invasion ability of the cells was assessed in vitro by using a transwell assay in accordance with the method detailed in our previous study [7]. Briefly, colon cancer cells (4.5 × 10^5^) transfected with si-LOC441461 or scrambled control were suspended in 2% FBS and seeded in the upper chamber of the transwells (Falcon, Corning Inc., Grand Island, NY, USA) with a coating of Matrigel (BD Biosciences, Franklin Lakes, NJ, USA) for the invasion assay or without the Matrigel coating for the migration assay. After incubation in a CO_2_ incubator at 37 °C for 12 or 24 h, the remaining cells in the upper chamber were removed using cotton swabs and the cells on the undersurface of the transwells were fixed with 10% formaldehyde solution. The cells on the undersurface of the transwells were stained with crystal violet solution, and the number of colon cancer cells was calculated through cell counting in three fields under a phase-contrast microscope. The invasion and migration abilities were quantified using the Ascent software (Thermo Scientific, Waltham, MA, USA). All the experiments were performed in triplicate.

### 4.15. Western Blotting

The cells were harvested 24 h after transient transfection, washed with PBS, and subjected to lysis in radioimmunoprecipitation assay buffer (50 mM Tris–HCl at a pH of 8.0, 150 mM NaCl, 1% NP40, 0.5% deoxycholate, and 0.1% sodium dodecyl sulfate) at 4 °C for 30 min. The lysed cells were collected and centrifuged to remove cell debris. Protein assays were performed using the Bio-Rad Protein Assay kit according to the Bradford dye-binding procedure (Bio-Rad, Hercules, CA, USA). Protein samples (40 μg) were separated using sodium dodecyl sulfate–polyacrylamide gel electrophoresis in 10% or 12% resolving gel by using a Mighty Small II Deluxe Mini Vertical Electrophoresis Unit (Hoefer, Inc., Holliston, MA, USA). Proteins were then electrotransferred to polyvinylidene difluoride membranes (PerkinElmer, Inc., Waltham, MA, USA) by using the Mighty Small Transfer Tank (Hoefer, Inc.). Subsequently, the membranes were blocked with a blocking buffer (50 mM Tris–HCl at a pH of 7.6, 150 mM NaCl, 0.1% Tween 20, 5% nonfat dried milk, and 0.05% sodium azide) for 1 h at room temperature and incubated overnight with the following primary antibodies at 4 °C: CCNA2 (1:1000; 18202-1-AP, Proteintech Group, Inc., Rosemont, IL, USA), CCNB1 (1:1000; 55004-1-AP, Proteintech Group, Inc.), CCND1 (1:1000; RM-9104-S, Thermo Fisher Scientific Inc.), CDK4 (1:1000; MS-299-P, Thermo Fisher Scientific Inc.), CDKN1B (1:1000; #3686, Cell Signaling Technology, Inc., Beverly, MA, USA), CDKN1A (1:1000; #2947, Cell Signaling Technology, Inc.), E-CAD (1:1000; GTX61329, GeneTex, Inc., Irvine, CA, USA), VIM (1:1000; GTX100619, GeneTex, Inc.), Twist1 (1:1000, GTX127310, GeneTex, Inc.), RhoA (1:1000; #2117, Cell Signaling Technology, Inc.), ROCK1 (1:1000; ab45171, Abcam, Cambridge, MA, USA), MLC (1:1000; #3672, Cell Signaling Technology, Inc.), pMLC (1:1000; #3671, Cell Signaling Technology, Inc.), LIMK1 (1:1000; #3842, Cell Signaling Technology, Inc.), p-LIMK1 (1:1000; #3841, Cell Signaling Technology, Inc.), and ACTB (1:2000; MAB1501, EMD Millipore, Billerica, MA, USA). The membranes were then incubated with anti-rabbit (sc-2004) or anti-mouse (sc-2005) immunoglobulin G horseradish-peroxidase-conjugated secondary antibodies (1:10000, Santa Cruz Biotechnology, Inc.) for 1 h at room temperature. After three washes with Tris-buffered saline containing Tween-20 buffer (50 mM Tris–HCl at a pH of 7.6, 150 mM NaCl, and 0.1% Tween-20), immunoreactive bands were detected using a WesternBright ECL substrate (Advansta, Menlo Park, CA, USA).

### 4.16. Statistical Analysis

The expression levels of LOC441461 in colon cancer tissues were analyzed using TCGA database or RT-PCR by employing Student’s *t* test. Cell proliferation, colony formation, soft agar assay, and migration–invasion experiments were performed in triplicate. The histograms present the mean values, and the error bars indicate the standard deviation. The difference was considered significant when *p* < 0.05.

## 5. Conclusions

We identified a novel lncRNA (i.e., LOC441461) that acted as an oncogene and had significantly increased expression in colon cancer. LOC441461 knockdown was noted to suppress colon cancer cell growth by impairing cell cycle progression. In addition, our findings established LOC441461 as a candidate as an adjuvant for chemotherapy in CRC.

## Figures and Tables

**Figure 1 cancers-12-03171-f001:**
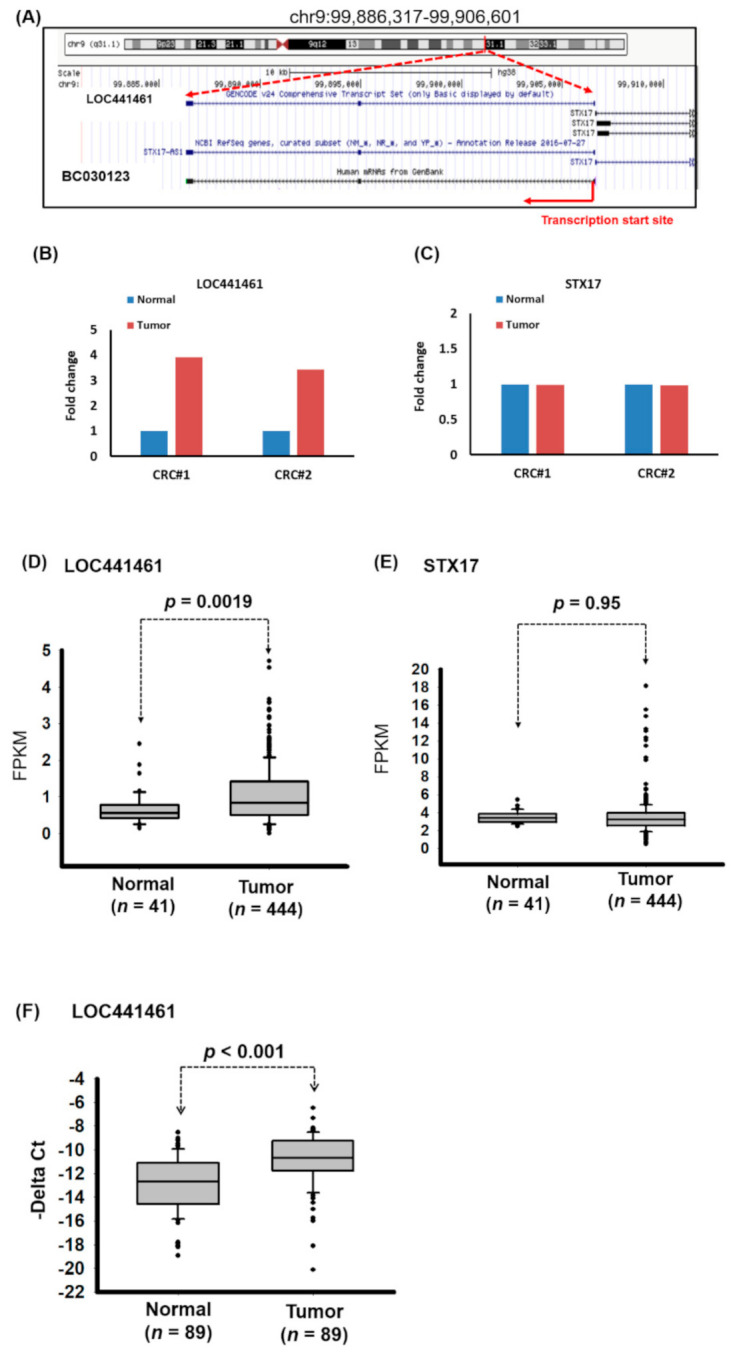
Abnormal expression of LOC441461 in human colorectal carcinoma (CRC). (**A**) Schematic representation of the location of LOC441461 in the human genome, as obtained from the website of the University of California, Santa Cruz (https://genome.ucsc.edu/). (**B**,**C**) Expression levels of LOC441461 and STX17 in the CRC samples and adjacent normal samples of two patients were determined using a microarray approach. (**D**,**E**) Expression levels of LOC441461 and STX17 were examined in human colorectal cancer samples obtained from The Cancer Genome Atlas (TCGA) database. Fragments per kilobase of transcripts per million was used to quantify the gene expression. (**F**) Expression levels of LOC441461 were examined using real-time (RT)-polymerase chain reaction (PCR) in CRC tissues and the corresponding normal tissues from 89 patients. The LOC441461 expression levels were statistically analyzed using Student’s *t* test. The difference was considered significant when *p* < 0.05.

**Figure 2 cancers-12-03171-f002:**
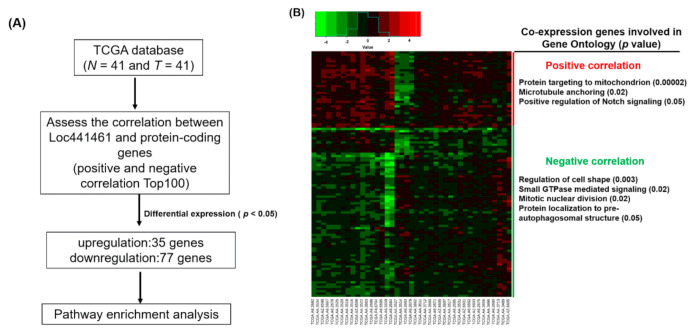
Identification of LOC441461-coexpressed genes through the TCGA database and pathway enrichment analysis. (**A**) Flowchart of identification of genes coexpressed with LOC441461 with significant differential expression (*p* < 0.05), as identified in CRC in the TCGA database. (**B**) Heat map of genes with significant expression (*p* < 0.05) in 41 CRC N-T pairs from TCGA database (left panel). The positively and negatively correlation genes were subjected to gene ontology analysis, and significantly involved pathways are displayed in the right panel.

**Figure 3 cancers-12-03171-f003:**
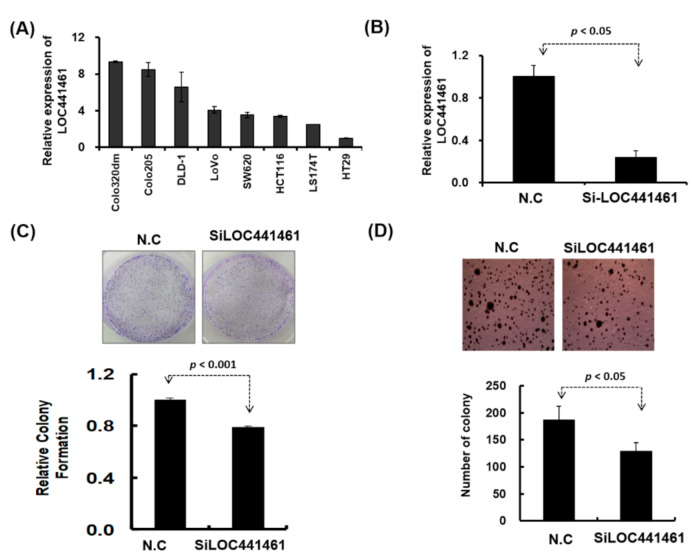
Biological function of LOC441461 was examined in colon cancer cells. (**A**) LOC441461 expression levels were examined in eight colon cancer cell lines by using RT-PCR. (**B**) LOC441461 expression levels were examined in SW620 cells with and without small interfering RNA (siRNA) transfection. (**C**) The colony formation assay was performed in the SW620 cells after transfection with si-Loc441461 and a scrambled control for 2 weeks. The cells were fixed and stained with crystal violet solution. The relative colony formation ability was then quantified (lower panel). (**D**) The anchorage-independent growth of SW620 cells with LOC441461 transfection was examined using the soft agar formation assay. A graph illustrates the quantified values. All experiments were performed in triplicate, and these data were analyzed using Student’s *t* test. The difference was considered significant when *p* < 0.05.

**Figure 4 cancers-12-03171-f004:**
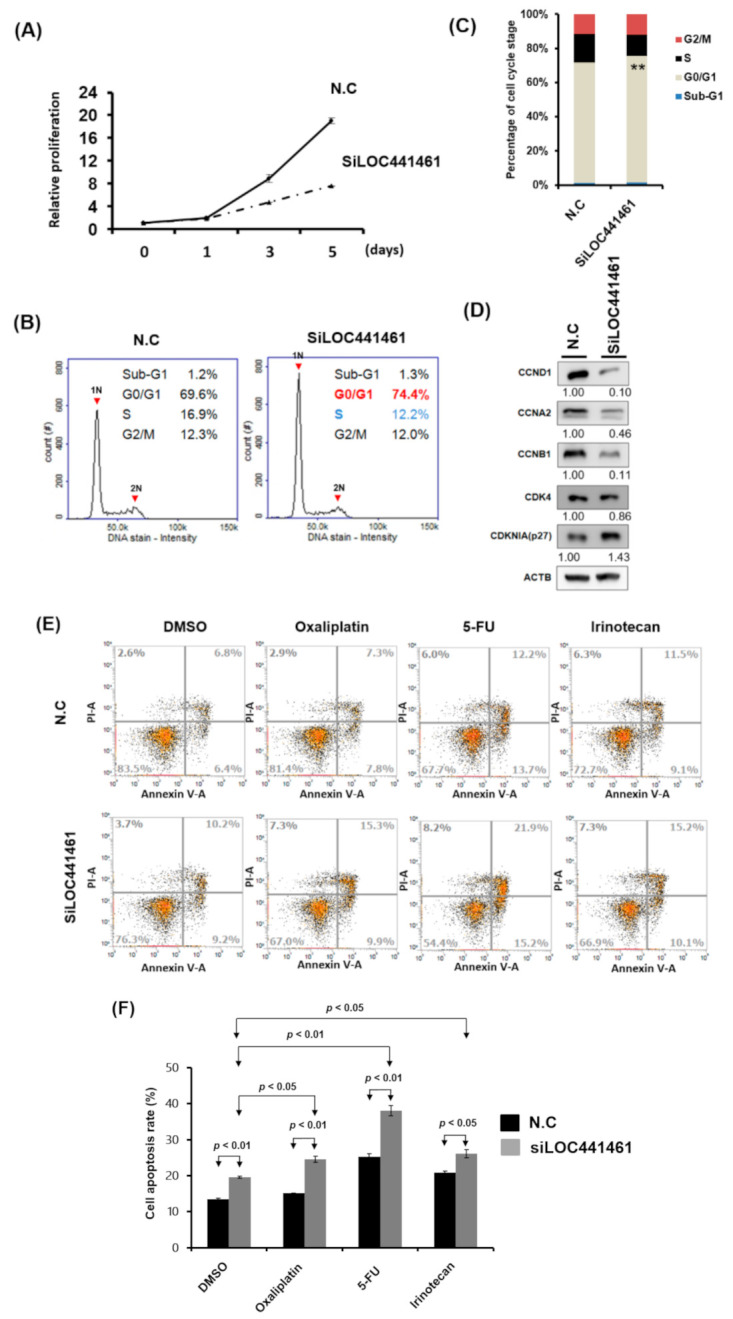
LOC441461 knockdown suppressed SW620 cell proliferation by inducing cell cycle arrest and apoptosis. (**A**) LOC441461 expression levels were knocked down with si-LOC441461 transfection in SW620 cells. Cell proliferation compared with the scrambled control was measured using the CellTiter-Glo One Solution assay at various time points (0, 1, 3, and 5 days). (**B**) Distribution of cells in three phases of the cell cycle was examined using the image flow cytometry assay. (**C**) Graph of each quantified phase. (**D**) Cell-cycle-related protein levels were examined using the Western blotting assay in SW620 cells with and without LOC441461 knockdown. (**E**,**F**) After exposure to 2 µg/mL oxaliplatin, 5-FU, or irinotecan for 48 h, cell apoptosis was examined using Annexin V assay, and the percentage of apoptosis was quantified in colon cancer cells with LOC441461 knockdown and in control cells. All the experiments were performed in triplicate, and these data were analyzed using Student’s *t* test. The difference was considered significant when *p* < 0.05. ** Indicates difference that is considered significant between the indicated groups *p* < 0.01.

**Figure 5 cancers-12-03171-f005:**
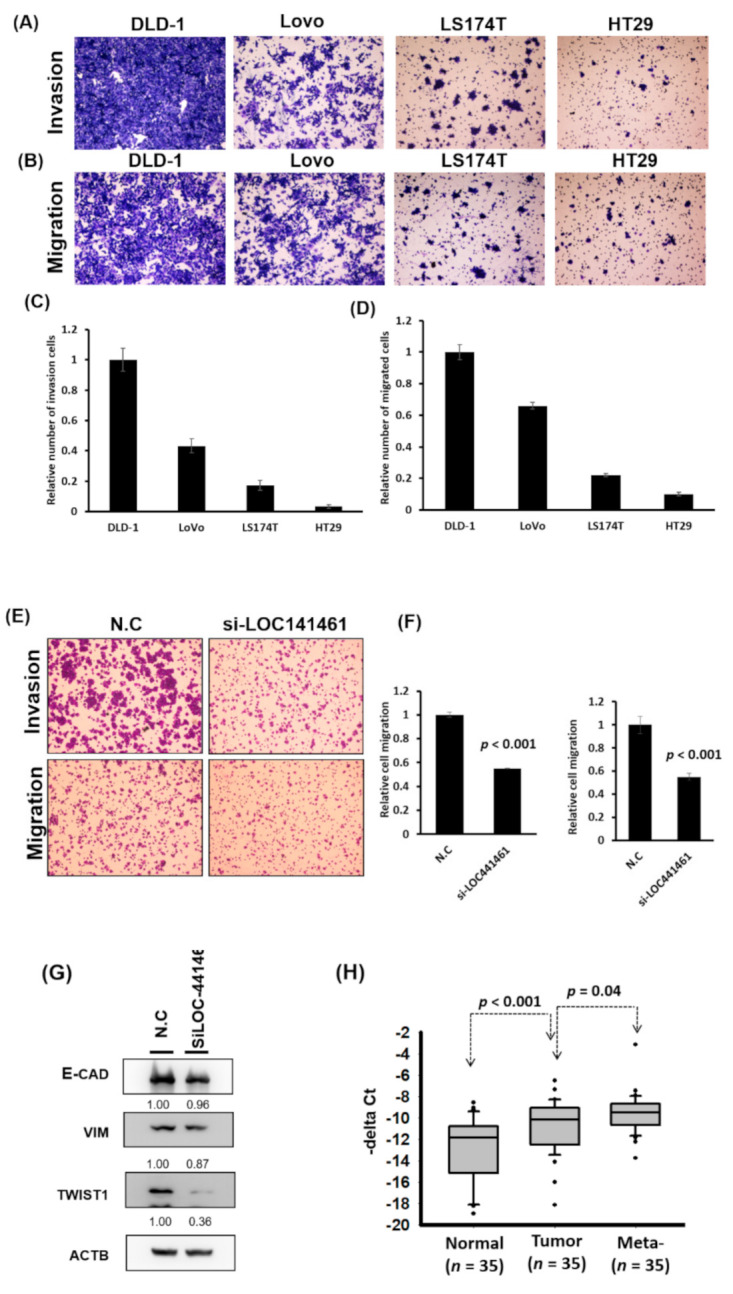
LOC441461 knockdown suppressed colon cancer cell motility. (**A**,**B**) Four colon cancer cell lines, DLD-1, LoVo, LS174T, and HT29, were seeded in the upper chamber of transwells with or without a coating of Matrigel (invasion and migration assay, respectively). After incubation for 12 h, the invading and migrating cells were counted. (**C**,**D**) The invasion and migration abilities were quantified using the Ascent software. (**E**) LOC441461 knockdown suppressed DLD-1 cell invasion and migration. (**F**) The invasion and migration abilities were further quantified using Ascent software. The migration and invasion experiments were performed in triplicate, and these data were analyzed using Student’s *t* test. The difference was considered significant when *p* < 0.05. (**G**) The expression levels of epithelial–mesenchymal transition markers were examined in the DLD-1 cell line with LOC441461 knockdown by using the Western blotting assay. (**H**) LOC441461 expression levels in the primary tumor, metastatic liver tumor, and corresponding normal mucosa of 35 patients with CRC were examined. LOC441461 expression levels in the CRC, adjacent normal mucosa, and liver metastases were analyzed using Student’s *t* test. (**I**) RhoA, ROCK, MLC, pMLC, LIMK1, and pLIMK1 protein levels in the DLD-1 cells with LOC441461 knockdown were examined using Western blotting. (**J**) Hypothetical mechanism through which LOC441461 is involved in the regulation of colon cancer growth and motility.

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
