# Peer review of "The Long Noncoding RNA LOC441461 (STX17-AS1) Modulates Colorectal Cancer Cell Growth and Motility"

_cancers, 2020, doi:10.3390/cancers12113171_

Round 1
Reviewer 1 Report
This new manuscript is a substantially modified version of the previous one. Significant and convincing amount of data have been added. Notably, the characterisation of LOC441461 knock-down have been performed in several colorectal cell lines, expressing different amount of the lncRNA. The amount of experiments supports the general conclusion and provides mechanistic insight into the potential role of LOC 441461.
I congratulate the authors for such an effort in performing the experiments and presenting so clearly the data.
I only have minor editing suggestions plus a question regarding Fig. 2.
Fig. 2: Total number of surviving patients after 4000 days should be identical in the two datasets since they analyse the same cohort of patients. Yet it seems that around 187 patients survived in panel A vs. 317 in panel B. Any explanation for this discrepancy?
Fig. 3: “upregulation and downregulation are somewhat misleading terms. I would rather recommend “negative or positive correlation” .
Fig 5B, and Supp. 2, 3, 4: cell cycle FACS profiles: adjust G1 and G2 peaks to N and 2N if possible. Are there the same number of acquired cells in each pair of graphs?
Supp. Fig6 : please annotate figures to make it instantly understandable which process is assayed in which cell line. Did the authors checked the efficacy of the KO in Ls174T cells, as compared to LoVo (this could explain the lack of observed effect, despite the fact that Ls174T express less LOC441461)?
Author Response
Dear Editor~
After reading the comments of Reviewer#3, we have removed the clinical part in this revised version, including Figure 2, Table 1 and Supplementary Table 1. We have also reordered the figures and tables and other information in the revised manuscript. In addition, we have provided a simple summary before the abstract. Furthermore, we have responded to the questions of the reviewers point by point. All changes were marked with yellow color in revised manuscript.
In page 1;
Simple Summary Section: Long noncoding RNA dysfunction is crucial for colorectal carcinoma (CRC) development. Whether the dysfunction of LOC441461, a novel lncRNA, can regulate cancer-related signaling pathways in cancer progression remains unclear. Here, we uncover the oncogenic role of LOC441461 in colon cancer cell growth and motility and identify a novel mechanism for LOC441461 knockdown-induced suppression of cancer motility through modulating RhoA/ROCK activity. This is the first report that LOC441461 knockdown impairs cell cycle progression and accelerates the apoptosis of colon cancer cells following chemotherapy drug treatment. The results suggest that LOC441461 expression confers drug sensitivity in colon cancer by inducing apoptosis. Our findings offer new insight into LOC441461 regulation and provide an application for colon cancer therapy in the future.
Reviewer#1
This new manuscript is a substantially modified version of the previous one. Significant and convincing amount of data have been added. Notably, the characterisation of LOC441461 knock-down have been performed in several colorectal cell lines, expressing different amount of the lncRNA. The amount of experiments supports the general conclusion and provides mechanistic insight into the potential role of LOC 441461. I congratulate the authors for such an effort in performing the experiments and presenting so clearly the data.
I only have minor editing suggestions plus a question regarding Fig. 2.
- 2:Total number of surviving patients after 4000 days should be identical in the two datasets since they analyse the same cohort of patients. Yet it seems that around 187 patients survived in panel A vs. 317 in panel B. Any explanation for this discrepancy?
Response: We apologize for this misinformation. Indeed, we analyzed the overall survival curve of LOC441461 and STX17 by using a dataset obtained from the TCGA database (444 patients with CRC). When we assessed the overall survival rate, we could observe a complete survival time for some patients, but others were still alive at the end of the tracking time. For these cases, we know only how long they survived. With these incomplete (censored) data, we analyzed the impacts of LOC441461 or STX17 expression on the survival curve of patients with CRC. The Y-axis should be defined as the cumulative survival rate, which may represent this result more accurately. According to Reviewer#3’s comments, we have removed the clinical impacts of LOC441461 and STX17 on colon cancer. Therefore, Figure 2 and Supplementary Table 1 have been removed from the revised version. Furthermore, the figures, tables, and descriptions have been revised in this version.
- 3: “upregulation and downregulation are somewhat misleading terms. I would rather recommend “negative or positive correlation”.
Response: We thank the reviewer for the suggestion, and we have made the revision.
- Fig 5B, and Supp. 2, 3, 4:cell cycle FACS profiles: adjust G1 and G2 peaks to N and 2N if possible. Are there the same number of acquired cells in each pair of graphs?
Response: We have labeled the G1 and G2 peaks to N and 2N in Figure 5B and Supplementary Figure 2, 3, and 4. In the study, 10 000 cells were subjected to cell cycle analysis for each pair of graphs.
- Fig6: please annotate figures to make it instantly understandable which process is assayed in which cell line. Did the authors checked the efficacy of the KO in Ls174T cells, as compared to LoVo (this could explain the lack of observed effect, despite the fact that Ls174T express less LOC441461)?
Response: We have annotated Supplementary Figure 6 to indicate which cells were used. The efficiency of the LOC441461 knockdown with siRNA is examined in Supplementary Figure 2 and 3. The knockdown efficiency was approximately 40% in both LS174T and Lovo cells after siRNA transfection. Therefore, we concluded that the inconsistent results may be related to differing genetic backgrounds of distinct colon cancer cells.
Reviewer#2
- English style is inadequate.
Reviewer#2′s response: The quality of English has been notably improved, but a more thorough revision is still required.
Response: We thank the reviewer for this suggestion. We have had our manuscript revised by academic editors.
- Abstract line 28: is it “higher” instead of lower? The LincRNA actually increases from normal, to primary CRC and to metastatic lesions.
Reviewer#2′s response: Correction OK.
- It is not clear when did the authors performed nuclear cytoplasmic localization, which is mentioned in line 212 of the Discussion.
Reviewer#2′s response:The authors have included a mention to the localisation of LOC441461 both in the Results and in the supplementary Fig. 1, but no mention of the method used for cellular fractionation is given. It must be included under Materials and Methods.
Response: We have provided the method for cellular fractionation in the supplementary data. ( in supplementary data, in page S11)
『Subcellular fractionation localization
The nuclear and cytosolic fractions were separated using the PARIS kit (Life Technologies) according to the manufacturer's instructions. Then, RNA was extracted using TRIZOL (Invitrogen), and 2 µg of total RNA was reverse-transcribed using random primers and SuperScript III Reverse Transcriptase (Invitrogen). Finally, RT-PCR was performed to determine the expression level of LOC441461. U6 was used as the nuclear marker, and GAPDH was used as the cytosolic fraction marker.
U6-F: 5′-CTCGCTTCGGCAGCACA-3′;
U6-R: 5′-AACGCTTCACGAATTTGCGT-3;
GAPDH-F: 5′- TGCACCACCAACTGCTTAGC-3′;
GAPDH-R: 5′- GGCATGGACTGTGGTCATGAG-3′;
LOC441461-F: 5′- TGATAAGCTGTTTAAACCAGAACCG-3′; and
LOC441461-R: 5′- GGCAACATTTCAGTTCCAGTG-3′.』
- The authors state that LOC441641 and STX1 share a bidirectional promoter. Is this an inference only gathered by genomic localization? Is there any evidence of promoter activity that can be obtained by analyzing Encode data? Please describe better the meaning of sharing a bidirectional promoter.
Reviewer#2′s response: The question has not been examined in detail, but, anyway, this is a minor point, which only tangentially affects the main body of the work.
Response: We attempted to resolve these concerns, and we apologize that the results do not address your misgivings. We are glad you are willing to accept this imperfect response.
- Regarding invasion ability of different CRC cell lines, only DLD-1 is shown for the high expressing group. Please show also the other cells mentioned in the text. It also should be commented that this is an indirect evidence of relevance of the LOC441641 to cell motility. Similarly, this same correlation should be done also for cell proliferation. What is the effect of LOC441641 siRNA on DLD1 cell growth and apoptosis?
Reviewer#2′s response: As requested by the Reviewer, the authors have added the results obtained with LoVo, and LS174T cell lines. Surprisingly, LOC441461 knockdown had no effect on the invasion and migration of LS174T cells. As the cell lines used differ in their genetic background, one can wonder whether the differences observed is due to off-target effects of the pooled siRNAs. Have the authors checked in any way that off-target effects are absent?
Response: We fully agree that the differences observed might result from off-target effects of the siRNA approach. Generally, reducing the concentration of a siRNA to a low effective dose where the intended target is still potently silenced can lead to a significant reduction in the number of off-targets that undergo significant changes in expression. In our study, we attempted to perform knockdown of LOC441461 with individual siRNA (10 nm) and pool siRNA (3.4 nm of siLOC441461_318, 3.3 nm of siLOC441461_384, and 3.3 nm of siLOC441461_432, respectively). We obtained similar results (Figure 4 vs. Supplementary Figure 1F and 1G). Low concentrations of individual siRNA in pooled siRNA can reduce the off-target effect; therefore, we performed all experiments using pooled siRNA. Of course, we cannot rule out the possibility of off-target effects in our results. However, we are more convinced that this result was caused by the different genetic backgrounds of distinct colon cancer cells.
- Shorter time points should be probably used for motility assays (maximum up to 12 h) to exclude that the migration/invasion inhibition is not due to growth arrest or apoptosis. In supplemental materials line 393 cell growth on other cell lines is described but this data is not discussed anywhere in the text.
Reviewer#2′s response: Satisfactorily answered.
- Please comment on why the effect on apoptosis is only visible by Annexin V staining and not by cell cycle analysis by FACS. Was the apoptosis evaluation performed following cytotoxic drug treatment? The Annexin V assay used is not mentioned in material and methods.
Reviewer#2′s response: The authors are right in their answer. The sub-G0/G1 population may include necrotic apart from apoptotic cells. The Annexin V assay has now been included under Materials and Methods.
- Is LOC441641 spliced? Where are the primes used to detect LOC441641 by qRT-PCR located? What about the primers used to detect STX17?
Reviewer#2′s response: The crux of the question has not been dealt with in the revised manuscript. The fact that only a band of about 500 bp be found in the experiment of supplementary Fig. 1B only assures that exon 2 is not skipped, but with the primers used for the PCR the skipping of exons 1 and/or 3 cannot be discarded. At any rate, this is also a minor point and a simple sentence to clear it may suffice, without need of further experiments.
Response: We attempted to resolve these concerns. We appreciate that you are willing to overlook this shortcoming.
- The Pathway gene enrichment analysis should be reinforced by in vitro evidences on LOC441641 silenced cells to further understand the mechanism of action of the LOC441641. It actually does not help in understanding the mechanism of action of LOC441641.
Reviewer#2′s response: The question has been aptly answered by adding further information in supplementary Figures 6 and 7. Although the meaning of arrows in supplementary Fig. 7B, it would be nice to mention it in the legend to the figure.
- Line 233-234: this statement should be revised; LOC441641 contributes to in vitro cancer cell proliferation and cell motility not to metastasis.
Reviewer#2′s response: Response OK.
Reviewer#3
The present study is divided between a rather larger pre-clinical fraction and a minor clinical part. Based on my experience the focus of my comments will adhere to the clinical part of the study.
Minor issue:
- In the abstract CRC is presented as the most common malignancy, which is not true. Please use the same phrases as in the introduction.
Response: We thank for reviewer for this reminder, and we have revised the text.
On page 1, at line 30:
Colorectal carcinoma (CRC) is 『one of』 the most prevalent cancer worldwide and has a high mortality rate.
Major issue:
- The clinical part of the study in general. This is poorly presented. If one are to interpret the clinical impact of these markers one need a very clear presentation of the patients. From the manuscript, we know that these patients were operated for CRC but that is more or less it. Clinical characteristics? Stage distribution? Treatment? In the supl Table 1, stages are presented revealing a highly heterogenous population.
Response: We fully agree with the reviewer’s comments. We should provide more detailed clinical information regarding the patients, including clinical characteristics and treatment. However, these data were obtained from a public database (TCGA), and it may be difficult to obtain more clinical information from the set. Indeed, these clinical impacts are of only minor value (and may even be misleading). Therefore, we have removed the clinical part in this revised version, including Figure 2, Table 1, and Supplementary Table 1.
- In addition, I simply cannot follow Figure 2. In A) OS, for the entire population is approximately 40% at 3000 days (the two curves end up more or less at the same spot). In B) that also presents OS data, for the exact same cohort outcome for the minor group (N=91) corresponds to the 40% but suddenly OS for the majority (N=353) is now around 80%??? These data does not add up. Did the authors suddenly censor a large amount of events (deaths) in the LOC441461 analyses – that for some reason was not called for in the STX17 analyses?
Response: We apologize for this misinformation. Indeed, we analyzed the overall survival curve of LOC441461 and STX17 using a dataset obtained from the TCGA database (444 patients with CRC). When we assessed the overall survival rate, we could observe a complete survival time for some patients, but others were still alive at the end of the tracking time. For these cases, we know only how long they survived. According to these incomplete (i.e., censored) data, we analyzed the impacts of LOC441461 or STX17 expression on the survival curve of patients with CRC. The Y-axis should be defined as the cumulative survival rate, which may represent this result more accurately.
Suggestion:
I would leave out the clinical part and focus this manuscript on all the pre-clinical tests, which seem sound and raises relevant hypotheses. The clinical part, as presented, has minor value and may actually worst case be misleading.
Response: We fully agree with this suggestion, and we have removed the clinical part from this version.

Reviewer 2 Report
The manuscript entitled “The Long Noncoding RNA LOC441461 (STX17-AS1) Modulates Colorectal Cancer Cell Growth and Motility” by Wang et al. describes a revised version of a previous manuscript submitted to Cancers. The present report analyses the answers given by the authors to the queries and observations of Reviewer #3.
- English style is inadequate.
The quality of English has been notably improved, but a more thorough revision is still required.
- Abstract line 28: is it “higher” instead of lower? The LincRNA actually increases from normal, to primary CRC and to metastatic lesions.
Correction OK.
- It is not clear when did the authors performed nuclear cytoplasmic localization, which is mentioned in line 212 of the Discussion.
The authors have included a mention to the localisation of LOC441461 both in the Results and in the supplementary Fig. 1, but no mention of the method used for cellular fractionation is given. It must be included under Materials and Methods.
- The authors state that LOC441641 and STX1 share a bidirectional promoter. Is this an inference only gathered by genomic localization? Is there any evidence of promoter activity that can be obtained by analyzing Encode data? Please describe better the meaning of sharing a bidirectional promoter.
The question has not been examined in detail, but, anyway, this is a minor point, which only tangentially affects the main body of the work.
- Regarding invasion ability of different CRC cell lines, only DLD-1 is shown for the high expressing group. Please show also the other cells mentioned in the text. It also should be commented that this is an indirect evidence of relevance of the LOC441641 to cell motility. Similarly, this same correlation should be done also for cell proliferation. What is the effect of LOC441641 siRNA on DLD1 cell growth and apoptosis?
As requested by the Reviewer, the authors have added the results obtained with LoVo, and LS174T cell lines. Surprisingly, LOC441461 knockdown had no effect on the invasion and migration of LS174T cells. As the cell lines used differ in their genetic background, one can wonder whether the differences observed is due to off-target effects of the pooled siRNAs. Have the authors checked in any way that off-target effects are absent?
- Shorter time points should be probably used for motility assays (maximum up to 12 h) to exclude that the migration/invasion inhibition is not due to growth arrest or apoptosis. In supplemental materials line 393 cell growth on other cell lines is described but this data is not discussed anywhere in the text.
Satisfactorily answered.
- Please comment on why the effect on apoptosis is only visible by Annexin V staining and not by cell cycle analysis by FACS. Was the apoptosis evaluation performed following cytotoxic drug treatment? The Annexin V assay used is not mentioned in material and methods.
The authors are right in their answer. The sub-G0/G1 population may include necrotic apart from apoptotic cells. The Annexin V assay has now been included under Materials and Methods.
- Is LOC441641 spliced? Where are the primes used to detect LOC441641 by qRT-PCR located? What about the primers used to detect STX17?
The crux of the question has not been dealt with in the revised manuscript. The fact that only a band of about 500 bp be found in the experiment of supplementary Fig. 1B only assures that exon 2 is not skipped, but with the primers used for the PCR the skipping of exons 1 and/or 3 cannot be discarded. At any rate, this is also a minor point and a simple sentence to clear it may suffice, without need of further experiments.
- The Pathway gene enrichment analysis should be reinforced by in vitro evidences on LOC441641 silenced cells to further understand the mechanism of action of the LOC441641. It actually does not help in understanding the mechanism of action of LOC441641.
The question has been aptly answered by adding further information in supplementary Figures 6 and 7. Although the meaning of arrows in supplementary Fig. 7B, it would be nice to mention it in the legend to the figure.
- Line 233-234: this statement should be revised; LOC441641 contributes to in vitro cancer cell proliferation and cell motility not to metastasis.
Response OK.
Author Response

(The authors gave the same response as above.)

Reviewer 3 Report
The present study is divided between a rather larger pre-clinical fraction and a minor clinical part. Based on my experience the focus of my comments will adhere to the clinical part of the study.
Minor issue:
- In the abstract CRC is presented as the most common malignancy, which is not true. Please use the same phrases as in the introduction.
Major issue:
- The clinical part of the study in general. This is poorly presented. If one are to interpret the clinical impact of these markers one need a very clear presentation of the patients. From the manuscript, we know that these patients were operated for CRC but that is more or less it. Clinical characteristics? Stage distribution? Treatment? In the supl Table 1, stages are presented revealing a highly heterogenous population.
- In addition, I simply cannot follow Figure 2. In A) OS, for the entire population is approximately 40% at 3000 days (the two curves end up more or less at the same spot). In B) that also presents OS data, for the exact same cohort outcome for the minor group (N=91) corresponds to the 40% but suddenly OS for the majority (N=353) is now around 80%??? These data does not add up. Did the authors suddenly censor a large amount of events (deaths) in the LOC441461 analyses – that for some reason was not called for in the STX17 analyses?
Suggestion:
I would leave out the clinical part and focus this manuscript on all the pre-clinical tests, which seem sound and raises relevant hypotheses. The clinical part, as presented, has minor value and may actually worst case be misleading.
Author Response

(The authors gave the same response as above.)

Round 2
Reviewer 3 Report
I find the answers satisfactory, and the current manuscript presents itself much more reliable now.
This manuscript is a resubmission of an earlier submission. The following is a list of the peer review reports and author responses from that submission.
Round 1
Reviewer 1 Report
In this manuscript, Wand and co-authors characterize LOC441461, a noncoding RNA of unknown function, in colorectal tumors. They show its expression correlates with lower survival. Expriments in three human colorectal cell lines show that down-regulation of LOC441461 expression reduces proflieration, migration and invasion through Matrigel.
It is an interesting finding, the data are convincing and the overall study is straight-forward. Figure legends are somewhat elusive so the manuscript would improve from adding experimental and statistical details, even if they can be found in the Material& Methods section (which researcher do not systematically consult).
MAJOR POINTS :
LOC441461 is a 563 bp lncRNA. qPCR and RNAseq from TGCA identify short sequence fragments. Author could perform classical RT-PCRs to ensure full-length expression of the 563-bp RNA.
Abstract: please correct
“Furthermore, significantly higher LOC expression occurred in primary colon tumors…”
Introduction: A non-coding gene is a group of genes encoding RNA transcripts…3”
2.1. ROC curve could be indicated as supplementary material, or an indication of cut-off value used. Table 1 is quite obscure to me. Also, the text refers to 480 patients but the total amounts to 445. Why?
2.2. Are there any link between the gene ontology analysis and the experimental data in Figure 4?
2.3. In Lovo and Ls174T: variation in G0/G1 population is minor and CDKNIA does not seem to increase.
Figure 5: part A, B, C, D were missing from the manuscript version.
Fig 6B: Explain the invasion assay briefly in the fig. legend, please.
In general, the manuscript would improve from more explanations, and a few more details about experiments and statistics in the figure legends.
MINOR POINTS:
Fig. 1A is too small. Is LOC441461 transcribed through the centromere? Please indicate where primers used for qPCR anneal to.
Fig. 1 B, C change presentation to compare expression in normal vs. tumor tissue for each sample
Fig. 1 D, E p= 0,0015 > 0,001 please correct text accordingly
Also: please improve annotation of figure and indicate the target
Table 1: n= 480 or 445?
Fig. 2A: What does “survimin” means exactly? What is the x axis unit : days? Months?
Fig. 3 C, B: enhance figure size
What is the difference between Fig. 3C and table S2? Could they be fused to one table/ figure?
Fig. 4C, D: Please indicate number of repetitions of biological experiments in the legend (here and in following figure legends). What does the statistics refer to?
Figure 4 D: How long does the experiment take place? Were siRNAs re-transfected during the 2-week experiment?
Fig.5 E is hardly distinguishable
Fig. 6A is a repetition of Fig. 4A
In general: please indicate in the figure legend the statistical test used.
Author Response
Reviewer #1
In this manuscript, Wang and co-authors characterize LOC441461, a noncoding RNA of unknown function, in colorectal tumors. They show its expression correlates with lower survival. Expriments in three human colorectal cell lines show that down-regulation of LOC441461 expression reduces proliferation, migration and invasion through Matrigel. It is an interesting finding, the data are convincing and the overall study is straight-forward. Figure legends are somewhat elusive so the manuscript would improve from adding experimental and statistical details, even if they can be found in the Material& Methods section (which researcher do not systematically consult).
MAJOR POINTS :
- LOC441461 is a 563 bp lncRNA. qPCR and RNAseq from TGCA identify short sequence fragments. Author could perform classical RT-PCRs to ensure full-length expression of the 563-bp RNA.
Response: We thank the reviewer for their suggestion. In the revised manuscript, we have indicated that paired primers were designed to amplify the full length of LOC441461 genes. As illustrated in Supplementary Figures 1A and 1B, PCR products with a length of ~500 bp were amplified using the LOC441461-2F and -2R primers. The PCR products were further subjected to confirmation by using the Sanger sequence.
In Results, page 5:
………………..(Figure 1A). 『We further determined the RNA transcript of LOC441461 using polymerase chain reaction (PCR) and Sanger sequencing approach. Our data indicated that the RNA transcript of LOC441461 consists with three exons (Supplementary Figure 1A and 1B).』Micorarray data………..
In Figure legend, page 38:
Supplementary Figure 1. LOC441461 knockdown suppressed colon cancer growth in SW620 cells.『 (A) Schema of the locations of four PCR primers used for LOC441461 amplification and three siRNA sequences designed for LOC441461 knockdown in this study. The gene structure of LOC441461 was obtained from the human genome database of the University of California, Santa Cruz. (B) The putative transcript of LOC441461 was identified using PCR with the LOC441461-2F/-2R primer in this study.』『(C)–(E) RNA expression levels of GAPDH, U6, and LOC441461 candidates, respectively, in the nucleus and cytoplasm were measured using RT-PCR. GAPDH acted as a marker for the cytoplasm, and U6 acted as a nuclear marker.』(F) After transfection with individual siRNA (si-LOC441461_318, si-LOC441461_384, and si-LOC441461_432), LOC441461 expression levels were examined using RT-PCR. (G) Cell proliferation was assessed in the SW620 cells with LOC441461 knockdown by using three individual siRNAs.
- Abstract: please correct “Furthermore, significantly higher LOC expression occurred in primary colon tumors…”
Response: We apologize for this mistake and have rectified it.
In the Abstract, page 2:
…………………. Furthermore, significantly 『higher』 LOC441461 expression was discovered in primary colon tumors and metastatic liver tumors than in the corresponding normal mucosa, and LOC441461 knockdown was noted to suppress colon cancer cell motility……………
- Introduction: A non-coding gene is a group of genes encoding RNA transcripts…3”
Response: We have rewritten this sentence.
In Introduction section, page 3:
『Noncoding RNAs (ncRNAs) are functional RNA transcripts』 that lack protein translation ability, and ncRNA dysfunction plays a crucial role in human cancer progression [4,5]……………….
- ROC curve could be indicated as supplementary material, or an indication of cut-off value used. Table 1 is quite obscure to me. Also, the text refers to 480 patients but the total amounts to 445. Why?
Response: We apologize for this unclear information. We downloaded all the transcriptome and clinical information data of 480 patients with colon cancer from TCGA. Survival information was lacking for some patients. Thus, after excluding the patients for whom no survival data were available, 444 patients were included in the survival analysis presented in our revised manuscript. We reanalyzed the effects of LOC441461 expression on clinical pathological features and the survival curve of colon cancer by using the data of the aforementioned 444 patients from TCGA database. We have added Supplementary Table 2 in the revised manuscript. In addition, we have added the optimal cutoff values in the manuscript.
In Result section, page 7:
…………………According to the defined cutoff values 『(3.38 for STX17 and 0.43 for LOC441461)』, the patients were separated into high and low STX17 or LOC441461 expression groups…………………………….
- Are there any link between the gene ontology analysis and the experimental data in Figure 4?
Response: We attempted to identify the corresponding biological pathway by using the coexpression approach. According to a previous study, coexpressed genes may have a similar biological role or coordinate function. Our gene ontology analysis revealed that LOC441461-coexpressed genes were involved in cell shape regulation, small-GTPase-mediated signaling transduction, and cell cycle regulation. Therefore, we examined the RhoA/ROCK signaling pathway in colon cancer cells with LOC441461 knockdown by using western blotting. Our data revealed that the RhoA/ROCK signaling activity as well as the cell membrane filopodium protrusions significantly decreased in DLD-1 cells with LOC441461 knockdown. We have illustrated this information in Figures 6I and 6J and Supplementary Figures 7A–7D. Some sentences have also been added to the Results section to describe our findings.
In Result section, page 15:
『Our results revealed that LOC441461 knockdown suppressed colon cancer cell growth and motility. According to pathway enrichment analysis, LOC441461-coexpressed genes were significantly involved in regulating the small GTPase activity, cell shape, and cell cycle. Studies have reported that the Rho family of small GTPases regulates crucial cellular processes, including cytoskeletal dynamics and cell migration and growth [26,27]. Therefore, we suggest that LOC441461 knockdown suppresses cancer cell growth and motility by blocking RhoA/ROCK signaling in colon cancer. We further examined the RhoA/ROCK/MLC2 and RhoA/ROCK/LIMK signaling in DLD-1 cells with LOC441461 knockdown. As illustrated in Figure 6I, the RhoA, ROCK and MLC expression were reduced in DLD-1 cells with LOC441461 knockdown. Our data also revealed that LOC441461 knockdown could suppress phosphorylation of MLC and LIMK1 (Figure 6I). Furthermore, LOC441461 knockdown decreased cell membrane filopodium protrusions in DLD-1 cells (Supplementary Figure 7A and 7B). In the aforementioned cells, the amount of G-actin (monomer) increased, whereas that of F-actin decreased (Supplementary Figure 7C and 7D).』 In summary, our study is the first to report the involvement of a novel oncogenic lncRNA, namely LOC441461, in colon cancer growth and cell motility 『through the modulation of the RhoA/ROCK signaling activity (Figure 6J).』
In Materials and Methods section, page 29-30:
Western blotting
The cells were harvested 24 hours after transient transfection, washed with PBS, and subjected to lysis in radioimmunoprecipitation assay buffer (50 mM Tris–HCl at a pH of 8.0, 150 mM NaCl, 1% NP40, 0.5% deoxycholate, and 0.1% sodium dodecyl sulfate) at 4 °C for 30 minutes. The lysed cells were collected and centrifuged to remove cell debris. Protein assays were performed using the Bio-Rad Protein Assay kit according to the Bradford dye-binding procedure (Bio-Rad). Protein samples (40 μg) were separated using sodium dodecyl sulfate–polyacrylamide gel electrophoresis in 10% or 12% resolving gel by using a Mighty Small II Deluxe Mini Vertical Electrophoresis Unit (Hoefer, Inc., Holliston, MA, USA). Proteins were then electrotransferred to polyvinylidene difluoride membranes (PerkinElmer, Inc., Waltham, MA, USA) by using the Mighty Small Transfer Tank (Hoefer, Inc.). Subsequently, the membranes were blocked with a blocking buffer (50 mM Tris–HCl at a pH of 7.6, 150 mM NaCl, 0.1% Tween 20, 5% nonfat dried milk, and 0.05% sodium azide) for 1 hour at room temperature and incubated overnight with the following primary antibodies at 4 °C: CCNA2 (1:1000; 18202-1-AP, Proteintech Group, Inc., Rosemont, IL, USA), CCNB1 (1:1000; 55004-1-AP, Proteintech Group, Inc.), CCND1 (1:1000; RM-9104-S, Thermo Fisher Scientific Inc.), CDK4 (1:1000; MS-299-P, Thermo Fisher Scientific Inc.), CDKN1B (1:1000; #3686, Cell Signaling Technology, Inc., Beverly, MA, USA), CDKN1A (1:1000; #2947, Cell Signaling Technology, Inc.), 『E-CAD (1:1000; GTX61329, GeneTex, Inc., Irvine, CA, USA), VIM (1:1000; GTX100619, GeneTex, Inc.), Twist1 (1:1000, GTX127310, GeneTex, Inc.), RhoA (1:1000; #2117, Cell Signaling Technology, Inc.), ROCK1 (1:1000; ab45171, Abcam.), MLC (1:1000; #3672, Cell Signaling Technology, Inc.), pMLC (1:1000; #3671, Cell Signaling Technology, Inc.), LIMK1 (1:1000; #3842, Cell Signaling Technology, Inc.), p-LIMK1 (1:1000; #3841, Cell Signaling Technology, Inc.),』 and ACTB (1:2000; MAB1501, EMD Millipore, Billerica, MA, USA). The membranes were then incubated with anti-rabbit (sc-2004) or anti-mouse (sc-2005) immunoglobulin G horseradish-peroxidase-conjugated secondary antibodies (1:10000, Santa Cruz Biotechnology, Inc.) for 1 hour at room temperature. After three washes with Tris-buffered saline containing Tween-20 buffer (50 mM Tris–HCl at a pH of 7.6, 150 mM NaCl, and 0.1% Tween-20), immunoreactive bands were detected using a WesternBright ECL substrate (Advansta, Menlo Park, CA, USA).
In Figure legends section, page 17:
Figure 6. LOC441461 knockdown suppressed colon cancer cell motility. (A) and (B) Four colon cancer cell lines, DLD-1, LoVo, LS174T, and HT29, were seeded in the upper chamber of transwells with or without a coating of Matrigel (invasion and migration assay, respectively). After incubation for 12 hours, the invading and migrating cells were counted. (C) and (D) The invasion and migration abilities were quantified using the Ascent software. (E) LOC441461 knockdown suppressed DLD-1 cell invasion and migration. (F) The invasion and migration abilities were further quantified using Ascent software. The migration and invasion experiments were performed in triplicate, and these data were analyzed using Student’s t test. The difference was considered significant when p < 0.05. (G) The expression levels of epithelial–mesenchymal transition markers were examined in the DLD-1 cell line with LOC441461 knockdown by using the Western blotting assay. (H) LOC441461 expression levels in the primary tumor, metastatic liver tumor, and corresponding normal mucosa of 35 patients with CRC were examined. LOC441461 expression levels in the CRC, adjacent normal mucosa, and liver metastases were analyzed using Student’s t test. 『(I) The RhoA, ROCK, MLC, pMLC, LIMK1 and pLIMK1 protein levels in the DLD-1 cells with LOC441461 knockdown were examined using Western blotting. (J) Hypothetical mechanism through which LOC441461 is involved in the regulation of colon cancer growth and motility.』
On page 44;
『Supplementary Figure 7. LOC441461 knockdown suppressed colon cancer cell motility through G-actin and F-actin polymeration. (A) DLD-1 cells treated with or without siLOC441461 siRNA were stained with rhodamine phalloidin, Alexa Fluor 488 DNase I conjugate, and 4′,6-diamidino-2-phenylindole for the detection of F-actin (red), G-actin (green), and nuclear (blue). (B) F-actin labeling with rhodamine phalloidin revealed that DLD-1 cells exhibited numerous filopodia, and the LOC441461-knockdown cells exhibited fewer filopodium fibers than the DLD-1 cells did. (C) and (D) The fluorescent intensity of G-actin and F-actin was calculated using a confocal microscope (n = 3). 』
- In Lovo and Ls174T: variation in G0/G1 population is minor and CDKNIA does not seem to increase.
Response: We completely agree with the reviewer’s comment that only a marginal change occurred in the G0/G1 population in the LoVo and Ls174T cells with LOC441461 knockdown. Because of the comment of reviewer #2, we further assessed the effects of LOC441461 knockdown on the growth and cell cycle of DLD-1 cells. Therefore, we have revised this description and have added some relevant text to the Discussion section.
In Results section; page 11-12:
…………………………Similar results of LOC441461-knockdown-induced colon cancer growth suppression were observed in the 『LoVo, LS174T, and DLD-1』 cell lines (Supplementary Figures 2–4). 『Moreover, the number of cells in the G0 and G1 phases was marginally higher in the LoVo, LS174T, and DLD-1 cell lines with LOC441461 knockdown than in the control cell line (Supplementary Figures 2–4).』
In the Discussion section, page 20:
『Our data indicated that LOC441461 knockdown suppressed colon cancer growth by inducing cell arrest in the G1 phase and providing drug sensitivity to SW620 cells. Our data indicated that LOC441461 knockdown induced cell cycle arrest in the G0/G1 phase. However, the numbers of cells in the G0 and G1 phases was only marginally higher in the LoVo, LS174T, and DLD-1 cell lines with LOC441461 knockdown than in the control group cell line (Supplementary Figures 2–4). Although LOC441461-knockdown-induced suppression of CCNB1 was observed in all colon cancer cells, the suppression of CCND1 and CDK4 was observed in SW620, DLD-1, and LS174T cells but not LoVo cells (Supplementary Figures 2–4). The marginally inconsistent aforementioned results may be related to the different genetic backgrounds of distinct colon cancer cells.』
- Figure 5: part A, B, C, D were missing from the manuscript version.
Response: These data and information are mentioned in the Results section, page 11.
- Fig 6B: Explain the invasion assay briefly in the fig. legend, please. In general, the manuscript would improve from more explanations, and a few more details about experiments and statistics in the figure legends.
Response: We thank the reviewer for their suggestion. We have provided more detailed information in the revised manuscript.
In the Figure Legend, page 17:
Figure 6. LOC441461 knockdown suppressed colon cancer cell motility. (A) and (B)『Four colon cancer cell lines, DLD-1, LoVo, LS174T, and HT29, were seeded in the upper chamber of transwells with or without a coating of Matrigel (invasion and migration assay, respectively). After incubation for 12 hours, the invading and migrating cells were counted. (C) and (D) The invasion and migration abilities were quantified using the Ascent software.』 (E) LOC441461 knockdown suppressed DLD-1 cell invasion and migration. (F) The invasion and migration abilities were further quantified using Ascent software. The migration and invasion experiments were performed in triplicate, and these data were analyzed using Student’s t test. The difference was considered significant when p < 0.05. (G) The expression levels of epithelial–mesenchymal transition markers were examined in the DLD-1 cell line with LOC441461 knockdown by using the Western blotting assay. (H) LOC441461 expression levels in the primary tumor, metastatic liver tumor, and corresponding normal mucosa of 35 patients with CRC were examined. 『LOC441461 expression levels in the CRC, adjacent normal mucosa, and liver metastases were analyzed using Student’s t test. (I) The RhoA, ROCK, MLC, pMLC, LIMK1 and pLIMK1 protein levels in the DLD-1 cells with LOC441461 knockdown were examined using Western blotting. (J) Hypothetical mechanism through which LOC441461 is involved in the regulation of colon cancer growth and motility.』
MINOR POINTS:
- 1A is too small. Is LOC441461 transcribed through the centromere? Please indicate where primers used for qPCR anneal to.
Response: We apologize for the poor resolution of Figure 1A. We have revised the figure and have clearly labeled the transcription start site of LOC441461 and gene location of LOC441461. Furthermore, we have labeled the locations of siRNAs and primers in Supplementary Figure 1A.
- 1 B, Cchange presentation to compare expression in normal vs. tumor tissue for each sample
Response: We have replaced Figures 1B and 1C with new figures.
- 1 D, Ep= 0,0015 > 0,001 please correct text accordingly. Also: please improve annotation of figure and indicate the target
Response: We have revised this text and have added clear annotations in the figure legends.
In result section, page 5;
…………….. We further examined the expression levels of LOC441461 and STX17 in colon cancer by analyzing The Cancer Genome Atlas (TCGA) database, which revealed that LOC441461 was significantly upregulated in colon cancer compared with adjacent normal tissues 『(p = 0.0019)』. By contrast, no difference was discovered in STX17 expression between colon cancer and normal tissues (p = 0.95; Figures 1D and 1E)…………..
Figure 1 legend, page 32:
Figure 1. Abnormal expression of LOC441461 in human colorectal carcinoma (CRC).
- Schematic representation of the location of LOC441461 in the human genome, as obtained from the website of the University of California, Santa Cruz (https://genome.ucsc.edu/). 『(B) and (C) Expression levels of LOC441461 and STX17 in the CRC samples and adjacent normal samples of two patients were determined using a microarray approach.』(D) and (E) Expression levels of LOC441461 and STX17 were examined in human colorectal cancer samples obtained from The Cancer Genome Atlas (TCGA) database. 『Fragments per kilobase of transcripts per million was used to quantify the gene expression.』 (F) Expression levels of LOC441461 were examined using real-time (RT)-polymerase chain reaction (PCR) in CRC tissues and the corresponding normal tissues from 89 patients.『 The LOC441461 expression levels were statistically analyzed using Student’s t The difference was considered significant when p < 0.05.』
- Table 1:n= 480 or 445?
Response: We downloaded the expression profiles and clinical data of 480 patients with CRC from TCGA database. Survival information was lacking for some patients. Thus, only 444 patients were included in the survival analysis presented in our manuscript. To prevent misunderstanding, we excluded the patients lacking survival data from the analysis. We reanalyzed the effects of LOC441461 expression on clinical pathological features and the survival curve of colon cancer by using data of the aforementioned 444 patients from TCGA database. We have added Figures 1D and E and Supplementary Table 1 in the revised manuscript.
In the Methods section, page 24:
Expression data from TCGA
Transcriptome expression data of colon cancer were downloaded from TCGA data portal (https://tcga-data.nci.nih.gov/tcga/dataAccessMatrix.htm). The expression profiles of 『444』 colon cancer tissues and 41 adjacent normal tissues were obtained from TCGA data portal. In this study, the transcriptome profiles of 444 patients with CRC were used to perform overall survival analysis by using the Kaplan–Meier method. In addition, the correlation between LOC441461 and protein-coding genes in colon cancer tissues from 41 patients was assessed using Pearson correlation. The 100 gene candidates with the strongest negative and positive correlations with LOC441461 were further examined in N-T paired colon cancer tissues from 41 patients, and the differentially expressed gene candidates in CRC were identified at the significance level p < 0.05.
In the Results section, page 7:
We analyzed TCGA dataset to understand the clinical prognostic value of LOC441461 in patients with colon cancer. Overall, the expression profiles and clinical information of 『444』patients with CRC were downloaded from TCGA………………….
- 2A: What does “survimin” means exactly? What is the x axis unit : days? Months?
Response: We have made the necessary revisions.
- 3 C, B:enhance figure size. What is the difference between Fig. 3C and table S2? Could they be fused to one table/ figure?
Response: We thank the reviewer for their suggestion. We have added a larger version of Figure 3B in the revised manuscript. In addition, we have deleted Figure 3C and retained Supplementary Table 2.
- 4C, D: Please indicate number of repetitions of biological experiments in the legend (here and in following figure legends). What does the statistics refer to?
Response: The experiments represented in this figure were performed in triplicate, and the data were analyzed using Student’s t test. The difference was considered significant when p < 0.05. We have provided this information in the Methods and materials section and the figure legends.
On page 13:
Figure 4. Biological function of LOC441461 was examined in colon cancer cells. (A) LOC441461 expression levels were examined in eight colon cancer cell lines by using RT-PCR. (B) LOC441461 expression levels were examined in SW620 cells with and without small interfering RNA (siRNA) transfection. (C) The colony formation assay was performed in the SW620 cells after transfection with si-Loc441461 and a scrambled control for 2 weeks. The cells were fixed and stained with crystal violet solution. The relative colony formation ability was then quantified (lower panel). (D) The anchorage-independent growth of SW620 cells with LOC441461 transfection was examined using the soft agar formation assay. A graph illustrates the quantified values. 『All experiments were performed in triplicate, and these data were analyzed using Student’s t test. The difference was considered significant when p < 0.05.』
- Figure 4 D: How long does the experiment take place? Were siRNAs re-transfected during the 2-week experiment?
Response: In this study, cells were knocked down through siRNA transfection and then cultured for 2 weeks. This experimental method is acceptable for examining the effects of genes on cell growth (the colony formation assay).
- 5 Eis hardly distinguishable
Response: LOC441461 knockdown only marginally induced cell apoptosis in colon cancer cells. Because of the comment of reviewer #3, we investigated the apoptosis effects of colon cancer cells with LOC441416 knockdown by using cytotoxic drug treatment. In the revised manuscript, we have assessed the effects of LOC441461 knockdown on cell apoptosis in SW620 cells treated with drugs. As illustrated in Figure 5E and 5F, LOC441461 knockdown accelerated the apoptosis of SW620 cells following oxaliplaton, 5-FU or irinotecan treatment. These results suggested that LOC441461 expression contributed to the drug sensitivity of colon cancer cells.
In result section, page 12:
『Our data also revealed that LOC441461 knockdown slightly induced colon cancer cell apoptosis (Figure 5E and 5F). We assessed whether LOC441461 expression contributed to drug responsiveness and examined the effects of LOC441461 knockdown on cell apoptosis in SW620 cells subjected to drug treatment. As depicted Figure 5E and 5F, LOC441461 knockdown accelerated the apoptosis of SW620 cells following oxaliplatin, 5-FU and irinotecan treatment. This result suggested that LOC441461 expression contributed to the drug sensitivity of colon cancer cells. 』
In Figure Legend, page 14;
Figure 5. LOC441461 knockdown suppressed SW620 cell proliferation by inducing cell cycle arrest and apoptosis
(A)LOC441461 expression levels were knocked down with si-LOC441461 transfection in SW620 cells. Cell proliferation compared with the scrambled control was measured using the CellTiter-Glo One Solution assay at various time points (0, 1, 3, and 5 days). (B) Distribution of cells in three phases of the cell cycle was examined using the image flow cytometry assay. (C) Graph of each quantified phase. (D) Cell-cycle-related protein levels were examined using the Western blotting assay in SW620 cells with and without LOC441461 knockdown. 『(E) and (F) After exposure to 2 µg/mL oxaliplatin, 5-FU, and irinotecan for 48 hours, the cell apoptosis was examined using Annexin V assay, and the percentage of apoptosis was quantified in colon cancer cells with LOC441461 knockdown and in control cells. All the experiments were performed in triplicate, and these data were analyzed using Student’s t test. The difference was considered significant when p < 0.05.』
- 6Ais a repetition of Fig. 4A
Response: This part has been removed. In addition, we have added the migration and invasion ability results for four colon cancer cells (Figures 6A–6D). We have also added some additional relevant text.
In the Results section, page 14:
……………………….. Notably, DLD-1 cells were more invasive than LoVo, LS174T, and HT29 cells, as illustrated in 『Figures 6A–6D』. The LOC441461 expression level was high in DLD-1 cells, moderate in LoVo and LS174T cells, and low in HT29 cells (Figure 4A). Furthermore, pathway enrichment analysis revealed that LOC441461-coexpressed genes were involved in microtubule anchoring and cell shape regulation (Figure 3B).……………
- In general: please indicate in the figure legend the statistical test used.
Response: We have added the method of statistical testing in the corresponding figure legend, including Fig. 1, 2, 4, 5, 6, supplementary Fig. 2., 3 and 4.

Reviewer 2 Report
The Manuscript by Jui-Ho Wang et al describes a lincRNA, named LOC441641, found upregulated in 2 CRC primary tumor samples compared with the corresponding normal counterpart using data from a previously published microarray by the same group.
In the present paper the authors start investigating LOC441641 biological functions and clinical significance using in vitro CRC cell lines and TCGA data.
The effects of in vitro LOC441641 down regulation are interesting, yet they are poorly developed and poorly presented.
Just to mention some weak points:
- English style is inadequate
- Abstract line 28: is it “higher” instead of lower? The LincRNA actually increases from normal, to primary CRC and to metastatic lesions.
- It is not clear when did the authors performed nuclear cytoplasmic localization, which is mentioned in line 212 of the Discussion.
- The authors state that LOC441641 and STX1 share a bidirectional promoter. Is this an inference only gathered by genomic localization? Are there any evidence of promoter activity that can be obtained by analyzing Encode data? Please describe better the meaning of sharing a bidirectional promoter.
- Regarding invasion ability of different CRC cell lines, only DLD-1 is shown for the high expressing group. Please show also the other cells mentioned in the text. It also should be commented that this is an indirect evidence of relevance of the LOC441641 to cell motility. Similarly, this same correlation should be done also for cell proliferation. What is the effect of LOC441641 siRNA on DLD1 cell growth and apoptosis?
- Shorter time points should be probably used for motility assays (maximum up to 12 h) to exclude that the migration/invasion inhibition is not due to growth arrest or apoptosis. In supplemental materials line 393 cell growth on other cell lines is described but this data is not discussed anywhere in the text.
- Please comment on why the effect on apoptosis is only visible by Annexin V staining and not by cell cycle analysis by FACS. Was the apoptosis evaluation performed following cytotoxic drug treatment? The Annexin V assay used is not mentioned in material and methods.
- Is LOC441641 spliced? Where are the primes used to detect LOC441641 by qRT-PCR located? What about the primers used to detect STX17?
- The Pathway gene enrichment analysis should be reinforced by in vitro evidences on LOC441641 silenced cells to further understand the mechanism of action of the LOC441641. It actually does not help in understanding the mechanism of action of LOC441641 this analyses.
- Line 233-234: this statement should be revised; LOC441641 contributes to in vitro cancer cell proliferation and cell motility not to metastasis.
Author Response
Reviewer #3
In the present paper the authors start investigating LOC441641 biological functions and clinical significance using in vitro CRC cell lines and TCGA data.
The effects of in vitro LOC441641 down regulation are interesting, yet they are poorly developed and poorly presented.
Just to mention some weak points:
- English style is inadequate
Response: The revised manuscript had been edited by a professional English editing company in revised version.
- Abstract line 28: is it “higher” instead of lower? The LincRNA actually increases from normal, to primary CRC and to metastatic lesions.
Response: We apologize for this mistake and have revised the text.
In Abstract section; page 2:
………………. Furthermore, significantly 『higher』LOC441461 expression was discovered in primary colon tumors and metastatic liver tumors than in the corresponding normal mucosa,……………
- It is not clear when did the authors performed nuclear cytoplasmic localization, which is mentioned in line 212 of the Discussion.
Response: We have added this data (Supplementary Figures 1C–1E). We have also added some sentences to the Results section (page 8).
In result section, page 11;
……………..level was high in colo320DM, colo205, and DLD-1; moderate in LoVo, SW620, HCT116, and LS174T; and low in HT29 cells, as illustrated in Figure 4A. 『Furthermore, we analyzed the subcellular localization of LOC441461, which revealed that LOC441461 expression occurred predominantly in the cytoplasm of colon cancer cells (Supplementary Figure 1C–1E).』 First, we examined the effect of LOC441461 on colon cancer cell growth by knocking down its expression in SW620 cells by using the small interfering RNA (siRNA) approach……….
In Legend of supplementary Figure, page 38;
Supplementary Figure 1. LOC441461 knockdown suppressed colon cancer growth in SW620 cells. (A) Schema of the locations of four PCR primers used for LOC441461 amplification and three siRNA sequences designed for LOC441461 knockdown in this study. The gene structure of LOC441461 was obtained from the human genome database of the University of California, Santa Cruz. (B) The putative transcript of LOC441461 was identified using PCR with the LOC441461-2F/-2R primer in this study. 『(C)–(E) RNA expression levels of GAPDH, U6, and LOC441461 candidates, respectively, in the nucleus and cytoplasm were measured using RT-PCR. GAPDH acted as a marker for the cytoplasm, and U6 acted as a nuclear marker.』 (F) After transfection with individual siRNA (si-LOC441461_318, si-LOC441461_384, and si-LOC441461_432), LOC441461 expression levels were examined using RT-PCR. (G) Cell proliferation was assessed in the SW620 cells with LOC441461 knockdown by using three individual siRNAs.
- The authors state that LOC441641 and STX1 share a bidirectional promoter. Is this an inference only gathered by genomic localization? Are there any evidence of promoter activity that can be obtained by analyzing Encode data? Please describe better the meaning of sharing a bidirectional promoter.
Response: We determined that LOC441461 and STX17 share a bidirectional promoter according to their genomic localization. We did not investigate whether this promoter had bidirectional activity. We evaluated the promoter activity of LOC441461 by using the Encode database and discovered that this region has potential to act as a promoter according to DNase I profiles and cis-regulatory elements. According to the response of reviewer #2 (minor point 21), STX17 and LOC441461 may be regulated through different mechanisms. In the present study, we focused on the biological function and clinical impact of LOC441461 dysfunction in colon cancer. Therefore, we did not examine or discuss whether the promoter has bidirectional activity.
In the Introduction section, page 3:
………………….Our previous study identified several dysregulated lncRNAs in CRC by using the microarray approach [7]. The biological function of LOC441461 in human cancer cells remains unclear. 『LOC441461 shares a bidirectional promoter with STX17 at human chromosome 9. Previous studies indicated that antisense ncRNA positively or negatively regulates the expression of sense protein-coding genes [22-25]. In this study, we assessed the expression levels of STX17 and LOC441461 in colon cancer. We determined that only LOC441461 was significantly overexpressed in colon cancer compared with adjacent normal tissues. Furthermore, our findings revealed that LOC441461 has a novel oncogenic role in regulating CRC cell growth and migration through modulating RhoA/ROCK signaling and can be a target for gene therapy.』
- Regarding invasion ability of different CRC cell lines, only DLD-1 is shown for the high expressing group. Please show also the other cells mentioned in the text. It also should be commented that this is an indirect evidence of relevance of the LOC441641 to cell motility. Similarly, this same correlation should be done also for cell proliferation. What is the effect of LOC441641 siRNA on DLD1 cell growth and apoptosis?
Response: In the revised manuscript, we examined the effects of LOC441461 knockdown on colon cancer motility in DLD-1, LoVo, and LS174T cells. Our data indicated that the invasion and migration abilities of DLD-1 and LoVo cells were significantly decreased after LOC441461 knockdown; however, LOC441461 knockdown had no effect on the invasion and migration abilities of LS174T cells. We also examined the effects of LOC441461 knockdown on the growth of DLD-1 cells and determined that cell growth was significantly decreased by impairing cell cycle progression. We have presented these new results in Supplementary Figures 4, 5 and 6 have added some sentences to the Results section.
In page 14-15:
…………………. The knockdown of LOC441461 expression significantly suppressed the invasion, migration, and wound healing abilities of DLD-1 cells (Figures 6E and 6F and Supplementary Figure 5). The Twist expression level in DLD-1 cells with LOC441461 knockdown was lower than that in the control 『cells (Figure 6G). We also examined the effects of LOC441461 knockdown on motility in LoVo and LS174T cells and discovered that LOC441461 knockdown suppressed the invasion and migration abilities of LoVo cells but not LS174T cells (Supplementary Figure 6).』 Furthermore, we examined the LOC441461 expression level in the adjacent normal tissues, primary CRC, and liver metastases of 35 patients with CRC. Our data indicated that LOC441641 expression was significantly higher (p < 0.001, Figure 6H)……………………….
Supplementary Figure 4, 5 and 6 legend In page 41-43:
『Supplementary Figure 4. LOC441461 knockdown suppressed DLD-1 cell growth by impairing cell cycle progression. (A) LOC441461 expression levels were examined using RT-PCR in DLD-1 cells with si-LOC441461 transfection. (B) Cell proliferation compared with the scrambled control was measured using the CellTiter-Glo One Solution assay at various time points (0, 1, 3, and 5 days). (C) The distribution of cells in three phases of the cell cycle was examined using the image flow cytometry assay. (D) Graph of each quantified phase. (E) Cell-cycle-related protein levels were examined using the Western blotting assay in DLD-1 cells with and without LOC441461 knockdown.
Supplementary Figure 5. LOC441461 knockdown suppressed DLD-1 cell wound healing ability. (A) The wound healing assay was performed in DLD-1 cells transfected with siLOC441461 and the scrambled control (N.C). (B) The relative migration ability was quantified by calculating the open wound length.
Supplementary Figure 6. Effects of LOC441461 knockdown on the invasion and migration abilities of LoVo and LS174T cells. (A) and (B) After the knockdown of LOC441461 in LoVo and LS174T cells, their invasion and migration abilities were assessed using the transwell assay. (C) and (D) The invasion and migration abilities were further quantified using the Ascent software. The difference was considered significant when p < 0.05.
』
- Shorter time points should be probably used for motility assays (maximum up to 12 h) to exclude that the migration/invasion inhibition is not due to growth arrest or apoptosis. In supplemental materials line 393 cell growth on other cell lines is described but this data is not discussed anywhere in the text.
Response: We completely agree with the reviewer’s comments. Therefore, we examined the effects of LOC441461 knockdown on cell migration by performing a wound healing assay. We further examined the migration ability of DLD-1 cells with LOC441461 knockdown at different time points. Our data indicated that short time periods (within 24 hours) should probably not be used for this cell migration assay. The wound had healed completely approximately 24 hours after being opened. In addition, FBS-free culture medium was used for the wound healing assay, which may have ruled out the possibility that the migration and invasion inhibition was caused due to growth suppression. We also observed similar cell densities for the control group and LOC441461 knockdown group. Overall, we suggest that LOC441461 knockdown may play a partial role in colon cancer cell motility. We have displayed data on the wound healing of DLD-1 cells with LOC441461 knockdown in Supplementary Figure 6 (Page 42).
- Please comment on why the effect on apoptosis is only visible by Annexin V staining and not by cell cycle analysis by FACS. Was the apoptosis evaluation performed following cytotoxic drug treatment? The Annexin V assay used is not mentioned in material and methods.
Response: The effect of LOC441461-knockdown-induced cell apoptosis was observed under Annexin V staining. Because this effect was weak, we could not observe an increased cell population at the sub-G1 stage in all colon cancer cells. We combined the LOC441461 knockdown and cytotoxic treatment and determined that LOC441461 knockdown enhanced the effect of cytotoxic-drug-induced apoptosis. We have presented this finding in the revised manuscript and have added some relevant sentences. In addition, we have described the Annexin V assay.
In the Results section, page 11-12;
Similar results of LOC441461-knockdown-induced colon cancer growth suppression were observed in the LoVo, LS174T, and DLD-1 cell lines (Supplementary Figures 2–4). 『Moreover, the number of cells in the G0 and G1 phases was marginally higher in the LoVo, LS174T, and DLD-1 cell lines with LOC441461 knockdown than in the control cell line (Supplementary Figures 2–4).
Our data also revealed that LOC441461 knockdown slightly induced colon cancer cell apoptosis (Figure 5E and 5F). We assessed whether LOC441461 expression contributed to drug responsiveness and examined the effects of LOC441461 knockdown on cell apoptosis in SW620 cells subjected to drug treatment. As depicted Figure 5E and 5F, LOC441461 knockdown accelerated the apoptosis of SW620 cells following oxaliplatin, 5-FU and irinotecan treatment. This result suggested that LOC441461 expression contributed to the drug sensitivity of colon cancer cells.』
In the Materials and methods section, page 27-28:
『Apoptotic cells stained with Annexin V, propidium iodide, and Hoechst 33342
Apoptotic cells were detected using the CF488A Annexin V and PI Apoptosis kit (Biotium, Fremont, CA, USA) and Hoechst 33342 (ChemoMetec). After staining, the cells were analyzed using an image flow cytometry assay (NucleoCounter NC-3000, ChemoMetec). NucleoView NC-3000 was then employed for automated image flow cytometry analysis.』
- Is LOC441641 spliced? Where are the primes used to detect LOC441641 by qRT-PCR located? What about the primers used to detect STX17?
Response: We amplified the full length of LOC441461 by using LOC441461 primers located at exon 1 and exon 3 (depicted in Supplementary Figures 1A and 1B). Our data indicated a major bands at about 500 bp and this PCR product was subjected for further sequencing.
In the Results section, page 5:
The role of one of these lncRNAs in CRC, namely LOC441461, remains unclear. Notably, LOC441461 is a 553-bp-long ncRNA that shares a bidirectional promoter with STX17 at human chromosome 9:99,886,317-99,906,601 (Figure 1A). 『We further determined the RNA transcript of LOC441461 using polymerase chain reaction (PCR) and Sanger sequencing approach. Our data indicated that the RNA transcript of LOC441461 consists with three exons (Supplementary Figure 1A and 1B).』 Microarray data revealed that LOC441461………………………………..
- The Pathway gene enrichment analysis should be reinforced by in vitro evidences on LOC441641 silenced cells to further understand the mechanism of action of the LOC441641. It actually does not help in understanding the mechanism of action of LOC441641 this analyses.
Response: We appreciate the reviewer's comments. Some experiments are actually still in progress. We attempted to examine the effect of Rho-ROCK signaling on the LOC441461-knockdown-induced inhibition of colon cancer cell motility and growth. Our data revealed that ROCK, pMLC and pLIMK1 expression as well as cell membrane filopodium protrusions were significantly decreased in DLD-1 cells with LOC441461 knockdown. However, the detailed mechanism through which LOC441461 regulates colon cancer cell growth and motility through RhoA/ROCK signaling remains unclear and must be investigated in the future. We have presented the relevant data in Figures 6I and 6J and Supplementary Figures 7A–7D and have added some sentences to describe our findings.
In Result section, page 15:
『Our results revealed that LOC441461 knockdown suppressed colon cancer cell growth and motility. According to pathway enrichment analysis, LOC441461-coexpressed genes were significantly involved in regulating the small GTPase activity, cell shape, and cell cycle. Studies have reported that the Rho family of small GTPases regulates crucial cellular processes, including cytoskeletal dynamics and cell migration and growth [26,27]. Therefore, we suggest that LOC441461 knockdown suppresses cancer cell growth and motility by blocking RhoA/ROCK signaling in colon cancer. We further examined the RhoA/ROCK/MLC2 and RhoA/ROCK/LIMK signaling in DLD-1 cells with LOC441461 knockdown. As illustrated in Figure 6I, the RhoA, ROCK and MLC expression were reduced in DLD-1 cells with LOC441461 knockdown. Our data also revealed that LOC441461 knockdown could suppress phosphorylation of MLC and LIMK1 (Figure 6I). Furthermore, LOC441461 knockdown decreased cell membrane filopodium protrusions in DLD-1 cells (Supplementary Figure 7A and 7B). In the aforementioned cells, the amount of G-actin (monomer) increased, whereas that of F-actin decreased (Supplementary Figure 7C and 7D).』 In summary, our study is the first to report the involvement of a novel oncogenic lncRNA, namely LOC441461, in colon cancer growth and cell motility 『through the modulation of the RhoA/ROCK signaling activity (Figure 6J).』
In discussion section, page 19-21;
『Pathway enrichment analysis revealed that LOC441461-coexpressed genes were significantly involved in regulating the small GTPase activity, cell shape, and cell cycle. In human malignancies, most Rho GTPases are aberrantly expressed and contribute to the regulation of cancer cell proliferation, metastasis, and angiogenesis. Knockdown of RhoA expression significantly suppressed cancer cell growth and tumorigenesis and enhanced the chemosensitivity of cancer cells to treatment with Adriamycin and 5-fluorouracil [38]. Zhang et al. reported that the blocking of the Rho-ROCK pathway impaired the cell cycle G1–S transition due to the increase in the P21(waf1/Cip1) and p27(Kip1) expression and decrease in the activities of CDK4 and CDK6[39]. Furthermore, cell-cycle-dependent Rho GTPase activity was shown to regulate cancer cell migration and invasion dynamically. A Rho-GTPase-activating protein, namely ARHGAP11A, was expressed in a cell-cycle-dependent manner and induced cell cycle arrest through interaction with p53 [40]. Interestingly, ARHGAP11A expression induced an increase in the relative Rac1 activity by blocking RhoA signaling, which led to an increase in the invasion ability of colon cancer cells [41]. Studies have demonstrated that microRNA and lncRNA modulate the growth and invasion properties of human cancer cells through the fine-tuning of RhoA/ROCK signaling [42-44]. Scholars have reported numerous lncRNAs involved in regulating human cancer cell growth and metastasis through the modulation of the RhoA pathway, including SNHG5, LOC554202, and MALAT1 [26,45]. In the present study, we discovered a novel oncogenic lncRNA (i.e., LOC441461) that regulated the growth and motility of colon cancer cells and conferred sensitivity through the RhoA-ROCK signaling activity. Our data indicated that LOC441461 knockdown suppressed colon cancer growth by inducing cell arrest in the G1 phase and providing drug sensitivity to SW620 cells. Our data indicated that LOC441461 knockdown induced cell cycle arrest in the G0/G1 phase. However, the numbers of cells in the G0 and G1 phases was only marginally higher in the LoVo, LS174T, and DLD-1 cell lines with LOC441461 knockdown than in the control group cell line (Supplementary Figures 2–4). Although LOC441461-knockdown-induced suppression of CCNB1 was observed in all colon cancer cells, the suppression of CCND1 and CDK4 was observed in SW620, DLD-1, and LS174T cells but not LoVo cells (Supplementary Figures 2–4). The marginally inconsistent aforementioned results may be related to the different genetic backgrounds of distinct colon cancer cells.』
In Materials and Methods section, page 28-30:
Western blotting
The cells were harvested 24 hours after transient transfection, washed with PBS, and subjected to lysis in radioimmunoprecipitation assay buffer (50 mM Tris–HCl at a pH of 8.0, 150 mM NaCl, 1% NP40, 0.5% deoxycholate, and 0.1% sodium dodecyl sulfate) at 4 °C for 30 minutes. The lysed cells were collected and centrifuged to remove cell debris. Protein assays were performed using the Bio-Rad Protein Assay kit according to the Bradford dye-binding procedure (Bio-Rad). Protein samples (40 μg) were separated using sodium dodecyl sulfate–polyacrylamide gel electrophoresis in 10% or 12% resolving gel by using a Mighty Small II Deluxe Mini Vertical Electrophoresis Unit (Hoefer, Inc., Holliston, MA, USA). Proteins were then electrotransferred to polyvinylidene difluoride membranes (PerkinElmer, Inc., Waltham, MA, USA) by using the Mighty Small Transfer Tank (Hoefer, Inc.). Subsequently, the membranes were blocked with a blocking buffer (50 mM Tris–HCl at a pH of 7.6, 150 mM NaCl, 0.1% Tween 20, 5% nonfat dried milk, and 0.05% sodium azide) for 1 hour at room temperature and incubated overnight with the following primary antibodies at 4 °C: CCNA2 (1:1000; 18202-1-AP, Proteintech Group, Inc., Rosemont, IL, USA), CCNB1 (1:1000; 55004-1-AP, Proteintech Group, Inc.), CCND1 (1:1000; RM-9104-S, Thermo Fisher Scientific Inc.), CDK4 (1:1000; MS-299-P, Thermo Fisher Scientific Inc.), CDKN1B (1:1000; #3686, Cell Signaling Technology, Inc., Beverly, MA, USA), CDKN1A (1:1000; #2947, Cell Signaling Technology, Inc.), 『E-CAD (1:1000; GTX61329, GeneTex, Inc., Irvine, CA, USA), VIM (1:1000; GTX100619, GeneTex, Inc.), Twist1 (1:1000, GTX127310, GeneTex, Inc.), RhoA (1:1000; #2117, Cell Signaling Technology, Inc.), ROCK1 (1:1000; ab45171, Abcam.), MLC (1:1000; #3672, Cell Signaling Technology, Inc.), pMLC (1:1000; #3671, Cell Signaling Technology, Inc.), LIMK1 (1:1000; #3842, Cell Signaling Technology, Inc.), p-LIMK1 (1:1000; #3841, Cell Signaling Technology, Inc.),』 and ACTB (1:2000; MAB1501, EMD Millipore, Billerica, MA, USA). The membranes were then incubated with anti-rabbit (sc-2004) or anti-mouse (sc-2005) immunoglobulin G horseradish-peroxidase-conjugated secondary antibodies (1:10000, Santa Cruz Biotechnology, Inc.) for 1 hour at room temperature. After three washes with Tris-buffered saline containing Tween-20 buffer (50 mM Tris–HCl at a pH of 7.6, 150 mM NaCl, and 0.1% Tween-20), immunoreactive bands were detected using a WesternBright ECL substrate (Advansta, Menlo Park, CA, USA).
In Figure legends section, page 17:
Figure 6. LOC441461 knockdown suppressed colon cancer cell motility. (A) and (B) Four colon cancer cell lines, DLD-1, LoVo, LS174T, and HT29, were seeded in the upper chamber of transwells with or without a coating of Matrigel (invasion and migration assay, respectively). After incubation for 12 hours, the invading and migrating cells were counted. (C) and (D) The invasion and migration abilities were quantified using the Ascent software. (E) LOC441461 knockdown suppressed DLD-1 cell invasion and migration. (F) The invasion and migration abilities were further quantified using Ascent software. The migration and invasion experiments were performed in triplicate, and these data were analyzed using Student’s t test. The difference was considered significant when p < 0.05. (G) The expression levels of epithelial–mesenchymal transition markers were examined in the DLD-1 cell line with LOC441461 knockdown by using the Western blotting assay. (H) LOC441461 expression levels in the primary tumor, metastatic liver tumor, and corresponding normal mucosa of 35 patients with CRC were examined. LOC441461 expression levels in the CRC, adjacent normal mucosa, and liver metastases were analyzed using Student’s t test. 『(I) The RhoA, ROCK, MLC, pMLC, LIMK1 and pLIMK1 protein levels in the DLD-1 cells with LOC441461 knockdown were examined using Western blotting. (J) Hypothetical mechanism through which LOC441461 is involved in the regulation of colon cancer growth and motility.』
On page 44;
『Supplementary Figure 7. LOC441461 knockdown suppressed colon cancer cell motility through G-actin and F-actin polymeration. (A) DLD-1 cells treated with or without siLOC441461 siRNA were stained with rhodamine phalloidin, Alexa Fluor 488 DNase I conjugate, and 4′,6-diamidino-2-phenylindole for the detection of F-actin (red), G-actin (green), and nuclear (blue). (B) F-actin labeling with rhodamine phalloidin revealed that DLD-1 cells exhibited numerous filopodia, and the LOC441461-knockdown cells exhibited fewer filopodium fibers than the DLD-1 cells did. (C) and (D) The fluorescent intensity of G-actin and F-actin was calculated using a confocal microscope (n = 3). 』
- Line 233-234: this statement should be revised; LOC441641 contributes to in vitro cancer cell proliferation and cell motility not to metastasis.
Response: We thank the reviewer for this suggestion. We have revised this sentence.
In the Discussion section, page 19;
……………………First, we determined that LOC441461 has an oncogenic role in promoting colon cancer 『proliferation and motility』. Then, we identified LOC441461-coexpressed genes by………………………

Reviewer 3 Report
The results of the presented study have potential for use in future clinical practice (e.g. could be applied for the prognosis ). Some of the results presented in the manuscript have already been disseminated (Kuo-Wang Tsai. Long non-coding RNA LOC441461 modulates colorectal cancer cell growth through inducing cell cycle arrest at G1 phases. J Cancer Sci Ther. Global Summit on Oncology & Cancer, May 25-27, 2017 Osaka, Japan). I found this manuscript interesting and valuable, however I have some comments for the authors as detailed below.
-Abstract - Lines 28-29 - Incorrect sentence - According to Fig. 6E and the information from the further part of the manuscript ( lines 195-198), NOT lower but higher LOC441461 expression occurred in primary colon tumors and metastatic liver tumors in comparison to the corresponding normal mucosa.
- Line 44 – The unclear sentence: „A non-protein coding gene is a group of RNA transcripts…”
- Lines 61-63 - Legend of Fig. 1 - „The expression levels of LOC441461 and STX17 were determined to be differentially expressed in the CRC samples compared with the corresponding adjacent normal samples …” - According to Fig. 1B and the information from the further part of the manuscript (line 73): „the expression levels of STX17 did not change”.
- Lines 63-64 - Legend of Fig. 1D and E – „(D) and (E) The expression levels of LOC441461 and STX17 were examined in human colorectal cancer obtained from TCGA database” - Misleading description, because it follows that the first graph refers to LOC441461, the second to STX17. It should be the opposite and authors should put appropriate markings on the charts.
- There is no subtitle for the first part of the results described. The authors are advised to inclcude such subtitle consistently as they did later in the Results section.
- Table 1 - Under the table there are explanations of unnecessary abbreviations (e.g. DSS, DFS), but they were not used in it. There is no explanation for the used „OS” abbreviation.
- The manuscript text (lines 85-86) contains information about 480 patients, while the Table 1 presents data for 444. Why? No explanation.
- Line 102 - Legend of Fig. 2 - It is worth specifying the information - expression of which factor was positively correlated and which factor was negatively correlated with poor patient survival.
- Fig. 2 - Unclear description of the X axis.
- Fig. 3 - Captions too small, illegible.
- line 116 - a typo in the word "upregulatied”
- Fig. 5C – a typo in the word „Percentagr”
- Line 181 - Legend of Fig. 5D - Since the Western blot method has been used, it may be better to accurately write „cell cycle-related protein levels" instead of „the expression levels of cell cycle-related genes…” Moreover, it is worth placing a graph which illustrates quantified values.
- Fig. 6A - The graph shows the level of LOC441461 expression in various cell lines, but the authors in the manuscript text (lines 185-186) refer to cell invasion.
- Fig 6B - In addition to the photos, it is worth placing a graph which illustrates quantified values.
- 186-187 - Invalid reference to figure. Repetition of the results already described in the previous part (lines 141-143).
- No consistency in writing the name of the gene " LOC441461" - sometimes in uppercase and sometimes lowercase.
- There is confusion in the manuscript about the effect of LOC441461 knockdown on cancer cell function. Did LOC441461 knockdown cause different effects on different lines? Line 144 - The authors only mentioned gene knockdown in SW620 cells: „…we examined the effects of LOC441461 on colon cancer cell growth by knocking down its expression in SW620 cells by using the siRNA approach.” Lines 163-164 -The authors wrote: „…LOC441461 knockdown-induced colon cancer growth suppression through impaired cell cycle progression was observed in LoVo, and LS174T cell lines (Supplementary Figures 2 and 3). Lines 191-192 - The authors described the effect of gene knockdown in a completely different line, this time - on cell invasion and migration: „knockdown of LOC441461 expression could significantly suppress invasion and migration ability in 191 DLD-1 cells”. Wouldn't it be a better way to compare the impact of LOC441461 knockdown on all activities studied (growth, migration invasion) in all lines?
- Fig. 6C- I advise you to add the word "migration" in the legend of the figure: „LOC441461 202 knockdown could suppress DLD-1 cell invasion and migration…”
- Fig. 6D - The manuscript text lacks a commentary describing the results presented in the figure. Only the result for Twist1 is mentioned. Since LOC441461 knockdown causes the decreased level of Twist1, why doesn't the E-CAD level increase (if normally Twist1 downregulate it)?
- Lines 211-213- add the appropriate references
- Lines 213-214- add the appropriate references
- The manuscript title reflects only a fraction of what was included in the manuscript.
- Lack of introduction / explanation of the purpose of some experiments, e.g. STX17 expression determination and its relation to patients survival (but one third of the short Discussion is devoted to this issue).
- I advise you to expand the Discussion section. Many aspects were omitted (e.g. about the cell cycle, the proteins associated with it and studied in the manuscript presented, aspects related to cell migration and invasion are not discussed).
- Materials and Methods - 4.5. Real-time polymerase chain reaction -no information on primers for STX17
- Materials and Methods - 4.5. Real-time polymerase chain reaction -no information on general
qPCR conditions applied
- Materials and Methods - 4.14. Western blotting - no information on antibodies to: E-CAD, VIM, Twist1. Instead, there is information about antibodies to proteins that have not been studied (CASP-1, E2F1)
Author Response
Reviewer#2
The results of the presented study have potential for use in future clinical practice (e.g. could be applied for the prognosis). Some of the results presented in the manuscript have already been disseminated (Kuo-Wang Tsai. Long non-coding RNA LOC441461 modulates colorectal cancer cell growth through inducing cell cycle arrest at G1 phases. J Cancer Sci Ther. Global Summit on Oncology & Cancer, May 25-27, 2017 Osaka, Japan). I found this manuscript interesting and valuable, however I have some comments for the authors as detailed below.
- -Abstract - Lines 28-29 - Incorrect sentence - According to Fig. 6E and the information from the further part of the manuscript ( lines 195-198), NOT lower but higher LOC441461 expression occurred in primary colon tumors and metastatic liver tumors in comparison to the corresponding normal mucosa.
Response: We apologize for this mistake and have rectified it.
In the Abstract, page 2:
…………………. Knockdown of the LOC441461 expression significantly suppressed colon cancer cell growth by impairing cell cycle progression and inducing cell apoptosis. Furthermore, significantly 『higher』LOC441461 expression was discovered in primary colon tumors and metastatic liver tumors than in the corresponding normal mucosa, and LOC441461 knockdown was noted to suppress colon cancer cell motility.……………
- - Line 44 – The unclear sentence: „A non-protein coding gene is a group of RNA transcripts…”
Response: We apologize for this unclear sentence and have revised it.
In Introduction section, page 3:
『Noncoding RNAs (ncRNAs) are functional RNA transcripts』 that lack protein translation ability, and ncRNA dysfunction plays a crucial role in human cancer progression [4,5]. …………….
- - Lines 61-63 - Legend of Fig. 1 - „The expression levels of LOC441461 and STX17 were determined to be differentially expressed in the CRC samples compared with the corresponding adjacent normal samples …” - According to Fig. 1B and the information from the further part of the manuscript (line 73): „the expression levels of STX17 did not change”.
Response: We apologize for this confusing sentence and have amended it.
In Figure Legend section, page 6-7;
Figure 1. Abnormal expression of LOC441461 in human colorectal carcinoma (CRC).
- Schematic representation of the location of LOC441461 in the human genome, as obtained from the website of the University of California, Santa Cruz (https://genome.ucsc.edu/). 『(B) and (C) Expression levels of LOC441461 and STX17 in the CRC samples and adjacent normal samples of two patients were determined using a microarray approach.』 (D) and (E) Expression levels of LOC441461 and STX17 were examined in human colorectal cancer samples obtained from The Cancer Genome Atlas (TCGA) database. Fragments per kilobase of transcripts per million was used to quantify the gene expression. (F) Expression levels of LOC441461 were examined using real-time (RT)-polymerase chain reaction (PCR) in CRC tissues and the corresponding normal tissues from 89 patients. The LOC441461 expression levels were statistically analyzed using Student’s t The difference was considered significant when p < 0.05.
- - Lines 63-64 - Legend of Fig. 1D and E – „(D) and (E) The expression levels of LOC441461 and STX17 were examined in human colorectal cancer obtained from TCGA database” - Misleading description, because it follows that the first graph refers to LOC441461, the second to STX17. It should be the opposite and authors should put appropriate markings on the charts.
Response: We apologize for this confusing sentence. The order of these data has been altered.
- - There is no subtitle for the first part of the results described. The authors are advised to inclcude such subtitle consistently as they did later in the Results section.
Response: We have removed the subtitle to the first part of the Results.
- - Table 1 - Under the table there are explanations of unnecessary abbreviations (e.g. DSS, DFS), but they were not used in it. There is no explanation for the used „OS” abbreviation.
Response: We have revised this part and have explained the abbreviation “OS.”
- - The manuscript text (lines 85-86) contains information about 480 patients, while the Table 1 presents data for 444. Why? No explanation.
Response: We downloaded the expression profiles and clinical data of 480 patients with CRC from TCGA database. Survival information was lacking for some patients. Thus, only 444 patients were included in the survival analysis presented in our manuscript. To prevent misunderstanding, we excluded the patients lacking survival data from the analysis. We reanalyzed the effects of LOC441461 expression on clinical pathological features and the survival curve of colon cancer by using data of the aforementioned 444 patients from TCGA database. We have added Figures 1D and E and Supplementary Table 1 in the revised manuscript.
In the Methods section, page 24:
Expression data from TCGA
Transcriptome expression data of colon cancer were downloaded from TCGA data portal (https://tcga-data.nci.nih.gov/tcga/dataAccessMatrix.htm). The expression profiles of 『444』 colon cancer tissues and 41 adjacent normal tissues were obtained from TCGA data portal. In this study, the transcriptome profiles of 444 patients with CRC were used to perform overall survival analysis by using the Kaplan–Meier method. In addition, the correlation between LOC441461 and protein-coding genes in colon cancer tissues from 41 patients was assessed using Pearson correlation. The 100 gene candidates with the strongest negative and positive correlations with LOC441461 were further examined in N-T paired colon cancer tissues from 41 patients, and the differentially expressed gene candidates in CRC were identified at the significance level p < 0.05.
In the Results section, page 7:
We analyzed TCGA dataset to understand the clinical prognostic value of LOC441461 in patients with colon cancer. Overall, the expression profiles and clinical information of 『444』patients with CRC were downloaded from TCGA………………….
- - Line 102 - Legend of Fig. 2 - It is worth specifying the information - expression of which factor was positively correlated and which factor was negatively correlated with poor patient survival.
Response: We thank the reviewer for their suggestion. We have made appropriate modifications to the relevant text.
On page 8;
Figure 2. Correlation of the STX17 and LOC441461 expression levels with the overall survival of patients with CRC. The expression levels and clinical data were obtained from 444 patients with CRC.
『(A) and (B) Low STX17 and high LOC441461 expression were significantly correlated with poor overall survival of patients with CRC, as determined using Kaplan–Meier survival analysis.』
- - Fig. 2 - Unclear description of the X axis.
Response: We have revised this part.
- - Fig. 3 - Captions too small, illegible.
Response: We thank you for this suggestion. We have increased the size of Figure 3B in the revised manuscript. In accordance with the suggestion of reviewer #1, Figure 3C has been deleted and Supplementary Table 2 has been retained.
- - line 116 - a typo in the word "upregulatied”
Response: We have revised this typo.
In the Results section, page 9;
………………As illustrated in Figure 3B, the 『upregulated』 coexpressed genes were significantly enriched in targeting mitochondria,……………
- - Fig. 5C – a typo in the word „Percentagr”
Response: We have revised this part (Page 13, figure 5C).
- - Line 181 - Legend of Fig. 5D - Since the Western blot method has been used, it may be better to accurately write „cell cycle-related protein levels" instead of „the expression levels of cell cycle-related genes…” Moreover, it is worth placing a graph which illustrates quantified values.
Response: Thank you for this suggestion. This part has been modified.
In the legend of Figure 5, page 14:
Figure 5. LOC441461 knockdown suppressed SW620 cell proliferation by inducing cell cycle arrest and apoptosis
(A)LOC441461 expression levels were knocked down with si-LOC441461 transfection in SW620 cells. Cell proliferation compared with the scrambled control was measured using the CellTiter-Glo One Solution assay at various time points (0, 1, 3, and 5 days). (B) Distribution of cells in three phases of the cell cycle was examined using the image flow cytometry assay. (C) Graph of each quantified phase. (D) 『Cell-cycle-related protein levels were 』examined using the Western blotting assay in SW620 cells with and without LOC441461 knockdown. (E) and (F) After exposure to 2 µg/mL oxaliplatin, 5-FU, and irinotecan for 48 hours, the cell apoptosis was examined using Annexin V assay, and the percentage of apoptosis was quantified in colon cancer cells with LOC441461 knockdown and in control cells. All the experiments were performed in triplicate, and these data were analyzed using Student’s t test. The difference was considered significant when p < 0.05.
- - Fig. 6A - The graph shows the level of LOC441461 expression in various cell lines, but the authors in the manuscript text (lines 185-186) refer to cell invasion. - Fig 6B - In addition to the photos, it is worth placing a graph which illustrates quantified values.
Response: In the revised manuscript, the duplicated data (Figure 6A duplicated Figure 4A) have been deleted. Furthermore, we have examined and quantified the invasion and migration abilities of four colon cancer cell lines: DLD-1, LoVo, LS174T, and HT29. We have displayed these new results in Figures 6A–6D and have modified the description of the results.
In the legend of Figure 6, page 16-17:
Figure 6. LOC441461 knockdown suppressed colon cancer cell motility. (A) and (B)『 Four colon cancer cell lines, DLD-1, LoVo, LS174T, and HT29, were seeded in the upper chamber of transwells with or without a coating of Matrigel (invasion and migration assay, respectively). After incubation for 12 hours, the invading and migrating cells were counted. (C) and (D) The invasion and migration abilities were quantified using the Ascent software.』 (E) LOC441461 knockdown suppressed DLD-1 cell invasion and migration. (F) The invasion and migration abilities were further quantified using Ascent software. The migration and invasion experiments were performed in triplicate, and these data were analyzed using Student’s t test. The difference was considered significant when p < 0.05. (G) The expression levels of epithelial–mesenchymal transition markers were examined in the DLD-1 cell line with LOC441461 knockdown by using the Western blotting assay. (H) LOC441461 expression levels in the primary tumor, metastatic liver tumor, and corresponding normal mucosa of 35 patients with CRC were examined. LOC441461 expression levels in the CRC, adjacent normal mucosa, and liver metastases were analyzed using Student’s t test. (I) The RhoA, ROCK, MLC, pMLC, LIMK1 and pLIMK1 protein levels in the DLD-1 cells with LOC441461 knockdown were examined using Western blotting. (J) Hypothetical mechanism through which LOC441461 is involved in the regulation of colon cancer growth and motility.
- - 186-187 - Invalid reference to figure. Repetition of the results already described in the previous part (lines 141-143).
Response: The data presented in Figure 6A duplicated those presented in Figure 4A. In the revised manuscript, we have presented the invasion ability of the DLD-1, LoVo, LS174T, and HT29 cell lines. This ability is quantified and illustrated in Figures 6A–6D. To avoid confusion, we have modified the relevant description in the revised manuscript.
In the Results section, page 14:
LOC441461 involved in colon cancer invasion ability
Notably, colo320DM, colo205, and DLD-1 cells were more invasive than Lovo, LS174T, and HT29, as illustrated in Figure 6A-6D. The expression levels of LOC441461 were high in colo320DM, colo205, and DLD-1; medium in LoVo and LS174T; and low in HT29 cells (Figure 4A).
- No consistency in writing the name of the gene " LOC441461" - sometimes in uppercase and sometimes lowercase.
Response: We have carefully checked the manuscript and have revised it, especially the figures.
- -There is confusion in the manuscript about the effect of LOC441461 knockdown on cancer cell function. Did LOC441461 knockdown cause different effects on different lines? Line 144 - The authors only mentioned gene knockdown in SW620 cells: „…we examined the effects of LOC441461 on colon cancer cell growth by knocking down its expression in SW620 cells by using the siRNA approach.” Lines 163-164 -The authors wrote: „…LOC441461 knockdown-induced colon cancer growth suppression through impaired cell cycle progression was observed in LoVo, and LS174T cell lines (Supplementary Figures 2 and 3). Lines 191-192 - The authors described the effect of gene knockdown in a completely different line, this time - on cell invasion and migration: „knockdown of LOC441461 expression could significantly suppress invasion and migration ability in 191 DLD-1 cells”. Wouldn't it be a better way to compare the impact of LOC441461 knockdown on all activities studied (growth, migration invasion) in all lines?
Response:
We apologize for this confusion. In this study, we examined the effects of LOC441461 knockdown on the growth of SW620, LS174T, LoVo, and DLD-1 cells. LOC441461 knockdown consistently suppressed the growth of SW620, LS174T, LoVo, and DLD-1 cells through cell cycle arrest. However, minor differences were observed in the efficiency of LOC441461-knockdown-induced suppression of growth and apoptosis for the different colon cancer cell lines. In addition, we assessed the influence of LOC441461 knockdown on the induction of cell apoptosis following cotreatment with a chemotherapy drug in revised version. We determined that LOC441461 knockdown confers drug sensitivity to SW620 cells. Interestingly, the inhibition of migration and invasion by LOC441461 knockdown was only observed in DLD-1 and LoVo cells. These slightly inconsistent results were obtained possibly due to the different genetic backgrounds of different colon cancer cell lines. We have presented these findings and additional relevant text in the revised manuscript.
In the Results section, page 11-12:
Similar results of LOC441461-knockdown-induced colon cancer growth suppression were observed in the 『LoVo, LS174T, and DLD-1 cell lines (Supplementary Figures 2–4). Moreover, the number of cells in the G0 and G1 phases was marginally higher in the LoVo, LS174T, and DLD-1 cell lines with LOC441461 knockdown than in the control cell line (Supplementary Figures 2–4).』
On page 14:
…………….. The Twist expression level in DLD-1 cells with LOC441461 knockdown was lower than that in the control cells (Figure 6G). 『We also examined the effects of LOC441461 knockdown on motility in LoVo and LS174T cells and discovered that LOC441461 knockdown suppressed the invasion and migration abilities of LoVo cells but not LS174T cells (Supplementary Figure 6)』…………………….
In the Discussion section, page 20:
『In the present study, we discovered a novel oncogenic lncRNA (i.e., LOC441461) that regulated the growth and motility of colon cancer cells and conferred sensitivity through the RhoA-ROCK signaling activity. Our data indicated that LOC441461 knockdown suppressed colon cancer growth by inducing cell arrest in the G1 phase and providing drug sensitivity to SW620 cells. Our data indicated that LOC441461 knockdown induced cell cycle arrest in the G0/G1 phase. However, the numbers of cells in the G0 and G1 phases was only marginally higher in the LoVo, LS174T, and DLD-1 cell lines with LOC441461 knockdown than in the control group cell line (Supplementary Figures 2–4). Although LOC441461-knockdown-induced suppression of CCNB1 was observed in all colon cancer cells, the suppression of CCND1 and CDK4 was observed in SW620, DLD-1, and LS174T cells but not LoVo cells (Supplementary Figures 2–4). The marginally inconsistent aforementioned results may be related to the different genetic backgrounds of distinct colon cancer cells.』
In Supplementary Figure 4 Legend, page 41;
『Supplementary Figure 4. LOC441461 knockdown suppressed DLD-1 cell growth by impairing cell cycle progression. (A) LOC441461 expression levels were examined using RT-PCR in DLD-1 cells with si-LOC441461 transfection. (B) Cell proliferation compared with the scrambled control was measured using the CellTiter-Glo One Solution assay at various time points (0, 1, 3, and 5 days). (C) The distribution of cells in three phases of the cell cycle was examined using the image flow cytometry assay. (D) Graph of each quantified phase. (E) Cell-cycle-related protein levels were examined using the Western blotting assay in DLD-1 cells with and without LOC441461 knockdown. All experiments were performed in triplicate, and these data were analyzed using Student’s t test. The difference was considered significant when p < 0.05.』
In Supplementary Figure 6 Legend, page 43;
『Supplementary Figure 6. Effects of LOC441461 knockdown on the invasion and migration abilities of LoVo and LS174T cells. (A) and (B) After the knockdown of LOC441461 in LoVo and LS174T cells, their invasion and migration abilities were assessed using the transwell assay. (C) and (D) The invasion and migration abilities were further quantified using the Ascent software. The difference was considered significant when p < 0.05. All experiments were performed in triplicate, and these data were analyzed using Student’s t test. The difference was considered significant when p < 0.05.』
- - Fig. 6C- I advise you to add the word "migration" in the legend of the figure: „LOC441461 202 knockdown could suppress DLD-1 cell invasion and migration…”
Response: Thank you for the suggestion. We have modified this part.
In Figure Legend of Figure 6C, page 17;
Figure 6. LOC441461 knockdown suppressed colon cancer cell motility. (A) and (B) Four colon cancer cell lines, DLD-1, LoVo, LS174T, and HT29, were seeded in the upper chamber of transwells with or without a coating of Matrigel (invasion and migration assay, respectively). After incubation for 12 hours, the invading and migrating cells were counted. (C) and (D) The invasion and migration abilities were quantified using the Ascent software. (E) LOC441461 knockdown suppressed DLD-1 cell invasion 『and migration.』 (F) The invasion and migration abilities were further quantified using Ascent software. The migration and invasion experiments were performed in triplicate, and these data were analyzed using Student’s t test. The difference was considered significant when p < 0.05. (G) The expression levels of epithelial–mesenchymal transition markers were examined in the DLD-1 cell line with LOC441461 knockdown by using the Western blotting assay. (H) LOC441461 expression levels in the primary tumor, metastatic liver tumor, and corresponding normal mucosa of 35 patients with CRC were examined. LOC441461 expression levels in the CRC, adjacent normal mucosa, and liver metastases were analyzed using Student’s t test. (I) The RhoA, ROCK, MLC, pMLC, LIMK1 and pLIMK1 protein levels in the DLD-1 cells with LOC441461 knockdown were examined using Western blotting. (J) Hypothetical mechanism through which LOC441461 is involved in the regulation of colon cancer growth and motility.
18.- Fig. 6D - The manuscript text lacks a commentary describing the results presented in the figure. Only the result for Twist1 is mentioned. Since LOC441461 knockdown causes the decreased level of Twist1, why doesn't the E-CAD level increase (if normally Twist1 downregulate it)?
Response: We completely agree with the reviewer’s comment that E-CAD expression should decrease under Twist upregulation in DLD-1 cells with LOC441461 knockdown. However, our data revealed no difference in the E-CAD expressions of DLD-1 cells with and without LOC441461 knockdown. LOC441461-knockdown-induced suppression of DLD-1 motility possibly occurred partially through the suppression of Twist expression. In the revised manuscript, we have presented new data to indicate that LOC441461 knockdown inhibited RhoA/ROCK signaling. Rho/ROCK activity induced the polymerization of G-actin into F-actin, which promoted cell motility; therefore, LOC441461 knockdown blocked F-actin polymerization and decreased cell membrane filopodium protrusions in DLD-1 cells. We have presented these new results in Figures 6I and 6J and Supplementary Figure 6.
In Result section, page 15:
『Our results revealed that LOC441461 knockdown suppressed colon cancer cell growth and motility. According to pathway enrichment analysis, LOC441461-coexpressed genes were significantly involved in regulating the small GTPase activity, cell shape, and cell cycle. Studies have reported that the Rho family of small GTPases regulates crucial cellular processes, including cytoskeletal dynamics and cell migration and growth [26,27]. Therefore, we suggest that LOC441461 knockdown suppresses cancer cell growth and motility by blocking RhoA/ROCK signaling in colon cancer. We further examined the RhoA/ROCK/MLC2 and RhoA/ROCK/LIMK signaling in DLD-1 cells with LOC441461 knockdown. As illustrated in Figure 6I, the RhoA, ROCK and MLC expression were reduced in DLD-1 cells with LOC441461 knockdown. Our data also revealed that LOC441461 knockdown could suppress phosphorylation of MLC and LIMK1 (Figure 6I). Furthermore, LOC441461 knockdown decreased cell membrane filopodium protrusions in DLD-1 cells (Supplementary Figure 7A and 7B). In the aforementioned cells, the amount of G-actin (monomer) increased, whereas that of F-actin decreased (Supplementary Figure 7C and 7D).』 In summary, our study is the first to report the involvement of a novel oncogenic lncRNA, namely LOC441461, in colon cancer growth and cell motility 『through the modulation of the RhoA/ROCK signaling activity (Figure 6J).』
In Materials and Methods section, page 29-30:
Western blotting
The cells were harvested 24 hours after transient transfection, washed with PBS, and subjected to lysis in radioimmunoprecipitation assay buffer (50 mM Tris–HCl at a pH of 8.0, 150 mM NaCl, 1% NP40, 0.5% deoxycholate, and 0.1% sodium dodecyl sulfate) at 4 °C for 30 minutes. The lysed cells were collected and centrifuged to remove cell debris. Protein assays were performed using the Bio-Rad Protein Assay kit according to the Bradford dye-binding procedure (Bio-Rad). Protein samples (40 μg) were separated using sodium dodecyl sulfate–polyacrylamide gel electrophoresis in 10% or 12% resolving gel by using a Mighty Small II Deluxe Mini Vertical Electrophoresis Unit (Hoefer, Inc., Holliston, MA, USA). Proteins were then electrotransferred to polyvinylidene difluoride membranes (PerkinElmer, Inc., Waltham, MA, USA) by using the Mighty Small Transfer Tank (Hoefer, Inc.). Subsequently, the membranes were blocked with a blocking buffer (50 mM Tris–HCl at a pH of 7.6, 150 mM NaCl, 0.1% Tween 20, 5% nonfat dried milk, and 0.05% sodium azide) for 1 hour at room temperature and incubated overnight with the following primary antibodies at 4 °C: CCNA2 (1:1000; 18202-1-AP, Proteintech Group, Inc., Rosemont, IL, USA), CCNB1 (1:1000; 55004-1-AP, Proteintech Group, Inc.), CCND1 (1:1000; RM-9104-S, Thermo Fisher Scientific Inc.), CDK4 (1:1000; MS-299-P, Thermo Fisher Scientific Inc.), CDKN1B (1:1000; #3686, Cell Signaling Technology, Inc., Beverly, MA, USA), CDKN1A (1:1000; #2947, Cell Signaling Technology, Inc.), 『E-CAD (1:1000; GTX61329, GeneTex, Inc., Irvine, CA, USA), VIM (1:1000; GTX100619, GeneTex, Inc.), Twist1 (1:1000, GTX127310, GeneTex, Inc.), RhoA (1:1000; #2117, Cell Signaling Technology, Inc.), ROCK1 (1:1000; ab45171, Abcam.), MLC (1:1000; #3672, Cell Signaling Technology, Inc.), pMLC (1:1000; #3671, Cell Signaling Technology, Inc.), LIMK1 (1:1000; #3842, Cell Signaling Technology, Inc.), p-LIMK1 (1:1000; #3841, Cell Signaling Technology, Inc.),』 and ACTB (1:2000; MAB1501, EMD Millipore, Billerica, MA, USA). The membranes were then incubated with anti-rabbit (sc-2004) or anti-mouse (sc-2005) immunoglobulin G horseradish-peroxidase-conjugated secondary antibodies (1:10000, Santa Cruz Biotechnology, Inc.) for 1 hour at room temperature. After three washes with Tris-buffered saline containing Tween-20 buffer (50 mM Tris–HCl at a pH of 7.6, 150 mM NaCl, and 0.1% Tween-20), immunoreactive bands were detected using a WesternBright ECL substrate (Advansta, Menlo Park, CA, USA).
In Figure legends section, page 17:
Figure 6. LOC441461 knockdown suppressed colon cancer cell motility. (A) and (B) Four colon cancer cell lines, DLD-1, LoVo, LS174T, and HT29, were seeded in the upper chamber of transwells with or without a coating of Matrigel (invasion and migration assay, respectively). After incubation for 12 hours, the invading and migrating cells were counted. (C) and (D) The invasion and migration abilities were quantified using the Ascent software. (E) LOC441461 knockdown suppressed DLD-1 cell invasion and migration. (F) The invasion and migration abilities were further quantified using Ascent software. The migration and invasion experiments were performed in triplicate, and these data were analyzed using Student’s t test. The difference was considered significant when p < 0.05. (G) The expression levels of epithelial–mesenchymal transition markers were examined in the DLD-1 cell line with LOC441461 knockdown by using the Western blotting assay. (H) LOC441461 expression levels in the primary tumor, metastatic liver tumor, and corresponding normal mucosa of 35 patients with CRC were examined. LOC441461 expression levels in the CRC, adjacent normal mucosa, and liver metastases were analyzed using Student’s t test. 『(I) The RhoA, ROCK, MLC, pMLC, LIMK1 and pLIMK1 protein levels in the DLD-1 cells with LOC441461 knockdown were examined using Western blotting. (J) Hypothetical mechanism through which LOC441461 is involved in the regulation of colon cancer growth and motility.』
On page 44;
『Supplementary Figure 7. LOC441461 knockdown suppressed colon cancer cell motility through G-actin and F-actin polymeration. (A) DLD-1 cells treated with or without siLOC441461 siRNA were stained with rhodamine phalloidin, Alexa Fluor 488 DNase I conjugate, and 4′,6-diamidino-2-phenylindole for the detection of F-actin (red), G-actin (green), and nuclear (blue). (B) F-actin labeling with rhodamine phalloidin revealed that DLD-1 cells exhibited numerous filopodia, and the LOC441461-knockdown cells exhibited fewer filopodium fibers than the DLD-1 cells did. (C) and (D) The fluorescent intensity of G-actin and F-actin was calculated using a confocal microscope (n = 3). 』
19.- Lines 211-213- add the appropriate references. Lines 213-214- add the appropriate references
Response: We have examined subcellular localization of LOC441461 and added this data in supplementary Figure 1C-1E.
In Result section , page 11;
………………『Furthermore, we analyzed the subcellular localization of LOC441461, which revealed that LOC441461 expression occurred predominantly in the cytoplasm of colon cancer cells (Supplementary Figure 1C–1E).』 First, we examined the effect of LOC441461 on colon cancer cell growth by knocking down its expression in SW620 cells by using the small interfering RNA (siRNA) approach.
In discussion section, page 18;
………………The subcellular localization of LOC441461 was also analyzed, revealing that higher expression of LOC441461 in the cytoplasm than in the nucleus 『(Supplementary Figure 1C-1E)』……………
- The manuscript title reflects only a fraction of what was included in the manuscript.
Response: We have modified our manuscript’s title. The new title is “The Long Noncoding RNA LOC441461 (STX17-AS1) Modulates Colorectal Cancer Cell Growth and Motility”
21 - Lack of introduction / explanation of the purpose of some experiments, e.g. STX17 expression determination and its relation to patients survival (but one third of the short Discussion is devoted to this issue).
Response: Our previous study revealed that the bidirectional promoter may be coexpressed in human cells (SOX21-AS1). Studies have also reported that antisense noncoding RNA positively or negatively regulates the expression of sense protein-coding gene expression. In this study, we examined the expression levels of STX17 and LOC441461 (STX17-AS1) in colon cancer and determined that only LOC441461 was significantly upregulated in colon cancer tissues compared with adjacent normal tissues. Furthermore, STX17 and LOC441461 expression were uncorrelated in colon cancer tissues (correlation: 0.008, data not shown). These results implied that STX17 and LOC441461 may be regulated through different mechanisms. In the present study, we focused on the biological function and clinical impact of LOC441461 dysfunction in colon cancer tissues. We have added some relevant sentences to the Introduction section.
In the Introduction section, page 3:
………………….Our previous study identified several dysregulated lncRNAs in CRC by using the microarray approach [7]. The biological function of LOC441461 in human cancer cells remains unclear. 『LOC441461 shares a bidirectional promoter with STX17 at human chromosome 9. Previous studies indicated that antisense ncRNA positively or negatively regulates the expression of sense protein-coding genes [22-25]. In this study, we assessed the expression levels of STX17 and LOC441461 in colon cancer. We determined that only LOC441461 was significantly overexpressed in colon cancer compared with adjacent normal tissues. Furthermore, our findings revealed that LOC441461 has a novel oncogenic role in regulating CRC cell growth and migration through modulating RhoA/ROCK signaling and can be a target for gene therapy.』
- - I advise you to expand the Discussion section. Many aspects were omitted (e.g. about the cell cycle, the proteins associated with it and studied in the manuscript presented, aspects related to cell migration and invasion are not discussed).
Response: We thank the reviewer for their suggestion. In the revised manuscript, we have included considerable new data in accordance with the reviewer’s comments. Therefore, we have added some sentences to discuss the effects of LOC441461 knockdown on the modulation of RhoA/ROCK signaling and the cell cycle.
In Discussion Section; page 19-21;
Discussion
『Pathway enrichment analysis revealed that LOC441461-coexpressed genes were significantly involved in regulating the small GTPase activity, cell shape, and cell cycle. In human malignancies, most Rho GTPases are aberrantly expressed and contribute to the regulation of cancer cell proliferation, metastasis, and angiogenesis. Knockdown of RhoA expression significantly suppressed cancer cell growth and tumorigenesis and enhanced the chemosensitivity of cancer cells to treatment with Adriamycin and 5-fluorouracil [38]. Zhang et al. reported that the blocking of the Rho-ROCK pathway impaired the cell cycle G1–S transition due to the increase in the P21(waf1/Cip1) and p27(Kip1) expression and decrease in the activities of CDK4 and CDK6[39]. Furthermore, cell-cycle-dependent Rho GTPase activity was shown to regulate cancer cell migration and invasion dynamically. A Rho-GTPase-activating protein, namely ARHGAP11A, was expressed in a cell-cycle-dependent manner and induced cell cycle arrest through interaction with p53 [40]. Interestingly, ARHGAP11A expression induced an increase in the relative Rac1 activity by blocking RhoA signaling, which led to an increase in the invasion ability of colon cancer cells [41]. Studies have demonstrated that microRNA and lncRNA modulate the growth and invasion properties of human cancer cells through the fine-tuning of RhoA/ROCK signaling [42-44]. Scholars have reported numerous lncRNAs involved in regulating human cancer cell growth and metastasis through the modulation of the RhoA pathway, including SNHG5, LOC554202, and MALAT1 [26,45]. In the present study, we discovered a novel oncogenic lncRNA (i.e., LOC441461) that regulated the growth and motility of colon cancer cells and conferred sensitivity through the RhoA-ROCK signaling activity. Our data indicated that LOC441461 knockdown suppressed colon cancer growth by inducing cell arrest in the G1 phase and providing drug sensitivity to SW620 cells. Our data indicated that LOC441461 knockdown induced cell cycle arrest in the G0/G1 phase. However, the numbers of cells in the G0 and G1 phases was only marginally higher in the LoVo, LS174T, and DLD-1 cell lines with LOC441461 knockdown than in the control group cell line (Supplementary Figures 2–4). Although LOC441461-knockdown-induced suppression of CCNB1 was observed in all colon cancer cells, the suppression of CCND1 and CDK4 was observed in SW620, DLD-1, and LS174T cells but not LoVo cells (Supplementary Figures 2–4). The marginally inconsistent aforementioned results may be related to the different genetic backgrounds of distinct colon cancer cells.』
- - Materials and Methods - 4.5. Real-time polymerase chain reaction -no information on primers for STX17
Response: In our study, the expression levels of STX17 were examined by analyzing TCGA database.
- - Materials and Methods - 4.5. Real-time polymerase chain reaction -no information on general qPCR conditions applied
Response: We have added details regarding real-time PCR.
In the Materials and methods section, page 24:
Real-time polymerase chain reaction
RT-PCR
The obtained cDNA was used for RT-PCR analysis with LOC441461- and glyceraldehyde 3-phosphate dehydrogenase (GAPDH)-specific primers, and gene expression was detected using the Fast SYBR Green Master Mix (Applied Biosystems; Thermo Fisher Scientific Inc.). 『RT-PCR was performed under the following conditions: incubation at 94 °C for 10 min, followed by 40 cycles of incubation at 94 °C for 15 s and at 60 °C for 32 s.』 Finally, the expression of LOC441461 was normalized to that of GAPDH (ΔCt = gene Ct – GAPDH Ct). The individual primers used in this study were as follows:
- - Materials and Methods - 4.14. Western blotting - no information on antibodies to: E-CAD, VIM, Twist1. Instead, there is information about antibodies to proteins that have not been studied (CASP-1, E2F1)
Response: We apologize for the absence of this information. In the revised manuscript, we have included information on E-CAD, VIM, and Twist1 and have deleted information on CASP1 and E2F1. In addition, we also added RhoA, ROCK, MLC, pMLC, LIMK1 and pLIMK1 in revised version.
In page 28-30;
Western blotting
The cells were harvested 24 hours after transient transfection, washed with PBS, and subjected to lysis in radioimmunoprecipitation assay buffer (50 mM Tris–HCl at a pH of 8.0, 150 mM NaCl, 1% NP40, 0.5% deoxycholate, and 0.1% sodium dodecyl sulfate) at 4 °C for 30 minutes. The lysed cells were collected and centrifuged to remove cell debris. Protein assays were performed using the Bio-Rad Protein Assay kit according to the Bradford dye-binding procedure (Bio-Rad). Protein samples (40 μg) were separated using sodium dodecyl sulfate–polyacrylamide gel electrophoresis in 10% or 12% resolving gel by using a Mighty Small II Deluxe Mini Vertical Electrophoresis Unit (Hoefer, Inc., Holliston, MA, USA). Proteins were then electrotransferred to polyvinylidene difluoride membranes (PerkinElmer, Inc., Waltham, MA, USA) by using the Mighty Small Transfer Tank (Hoefer, Inc.). Subsequently, the membranes were blocked with a blocking buffer (50 mM Tris–HCl at a pH of 7.6, 150 mM NaCl, 0.1% Tween 20, 5% nonfat dried milk, and 0.05% sodium azide) for 1 hour at room temperature and incubated overnight with the following primary antibodies at 4 °C: CCNA2 (1:1000; 18202-1-AP, Proteintech Group, Inc., Rosemont, IL, USA), CCNB1 (1:1000; 55004-1-AP, Proteintech Group, Inc.), CCND1 (1:1000; RM-9104-S, Thermo Fisher Scientific Inc.), CDK4 (1:1000; MS-299-P, Thermo Fisher Scientific Inc.), CDKN1B (1:1000; #3686, Cell Signaling Technology, Inc., Beverly, MA, USA), CDKN1A (1:1000; #2947, Cell Signaling Technology, Inc.), 『E-CAD (1:1000; GTX61329, GeneTex, Inc., Irvine, CA, USA), VIM (1:1000; GTX100619, GeneTex, Inc.), Twist1 (1:1000, GTX127310, GeneTex, Inc.), RhoA (1:1000; #2117, Cell Signaling Technology, Inc.), ROCK1 (1:1000; ab45171, Abcam.), MLC (1:1000; #3672, Cell Signaling Technology, Inc.), pMLC (1:1000; #3671, Cell Signaling Technology, Inc.), LIMK1 (1:1000; #3842, Cell Signaling Technology, Inc.), p-LIMK1 (1:1000; #3841, Cell Signaling Technology, Inc.),』 and ACTB (1:2000; MAB1501, EMD Millipore, Billerica, MA, USA). The membranes were then incubated with anti-rabbit (sc-2004) or anti-mouse (sc-2005) immunoglobulin G horseradish-peroxidase-conjugated secondary antibodies (1:10000, Santa Cruz Biotechnology, Inc.) for 1 hour at room temperature. After three washes with Tris-buffered saline containing Tween-20 buffer (50 mM Tris–HCl at a pH of 7.6, 150 mM NaCl, and 0.1% Tween-20), immunoreactive bands were detected using a WesternBright ECL substrate (Advansta, Menlo Park, CA, USA).
